# The Volcanoes of Naples: how effectively mitigating the highest volcanic risk in the World?

Giuseppe De Natale[1,2], Claudia Troise[1,2], Renato Somma[1,3]

[1]Istituto Nazionale di Geofisica e Vulcanologia, Via Diocleziano 328, 80124 Naples, Italy
[2]CNR-INO, Via Campi Flegrei 34, 80078 Pozzuoli, Italy
[3]CNR-IRISS, Via Guglielmo Sanfelice, 8, 80134 Naples, Italy

*Correspondence to*: Giuseppe De Natale (Giuseppe.denatale@ingv.it)

**Abstract.** The Naples (Southern Italy) area has the highest volcanic risk in the World, due to the coexistence of three highly explosive volcanoes (Vesuvius, Campi Flegrei and Ischia) with extremely dense urbanisation. More than three millions people live to within twenty kilometres from a possible eruptive vent. Mitigating such an extreme risk is made difficult because volcanic eruptions forecast is today an empirical procedure with very uncertain outcome. This paper starts recalling the state of the art of eruption forecast, and then describes the main hazards in the Neapolitan area, shortly presenting the
activity and present state of its volcanoes. Then, it proceeds to suggest the most effective procedures to mitigate the extreme volcanic and associated risks. The problem is afforded in a highly multidisciplinary way, taking into account the main economic, sociological and urban issues. The proposed mitigation actions are then compared with the existing emergency plans, developed by Italian Civil Protection, by highlighting their numerous, very evident faults. Our study, besides regarding the most complex and extreme situation of volcanic risk in the World, gives guidelines to assessing and managing
volcanic risk in any densely urbanised area.

## 1. Introduction

Volcanic eruptions, in particular super-eruptions from large calderas, represent one of the highest natural threats to the mankind (Newhall and Dzurisin 1988, Papale and Marzocchi, 2019). However, the eruption forecast is still an empirical, largely uncertain practice. The successful forecasts are very few, and sometimes obtained too shortly before the eruption
(Consensus study report, 2017). Globally, the percentage of timely and correct volcanic alarms is still very low, and the failure cases often resulted in considerable human losses (Winson et al., 2014). Many volcanoes, on the Earth, are located in remote areas, not densely populated. In remote areas, the only threatening volcanoes are the very explosive ones, in particular the large collapse calderas, which are able to produce large scale catastrophes. However, there are places on the Earth in which very explosive volcanic areas are also densely populated; in such cases, the volcanic threat can be very high
also in case of moderate to small eruptions, and the volcanic risk can attain extremely high values. This is exactly the case of the volcanoes located in the Neapolitan area (Southern Italy). The Neapolitan volcanism is hosted in the larger Campania Plain, a former volcanic field which, since about 300.000 years to 39.000 years BP, gave rise to six Ignimbritic eruptions; the last one (Campanian Ignimbrite, 39.000 y BP) has been the largest eruption ever occurred in Europe, at our knowledge. The Neapolitan volcanic area has always been (since 4000 years at least) one of the most densely populated ones in the World
(Mastrolorenzo et al., 2006a), and one of the main cradles of the Western civilization (Astarita, 2013). The high attractions of this area, namely mild climate, fertile land, landscapes, hot springs and natural bays and inlets, have been in fact always perceived as largely overcoming the volcanic threat. Neapolitan volcanoes are among the most known in the World, since ancient times: Vesuvius, Campi Flegrei and Ischia island; all of them very explosive volcanoes. Ischia island has been, in the VIII century BC, the first Greek colony in Italian peninsula, named at the time: 'Magna Grecia' (Carratelli, 1985). The large
metropolitan area of Naples, the largest one in Southern Italy, contains about three millions people, all of them living to

within about 20 km from a possible eruptive vent. The Neapolitan area is then characterized by the highest volcanic risk in the World, which must be someway managed and mitigated. Actually, there are emergency plans, issued by the National Department of Civil Protection in Italy, for the two main volcanic areas: Vesuvius and Campi Flegrei. This paper describes the main volcanological issues of the three Neapolitan volcanic areas, the associated risks, and describes the main problems of the present emergency plans. It then proceeds to define the most important features that a realistic emergency plan, aimed to effectively mitigate the extreme volcanic risk of the area, should contain. Mitigation of volcanic risk is here afforded in a highly multidisciplinary framework, because the main actions needed start from volcanological considerations, but necessarily involve economic, sociologic and urbanistic considerations. The extreme risk of Neapolitan volcanic area is a paradigm for other volcanic areas of the World. Addressing the best procedures to mitigate volcanic risk here is a very important bench mark, for any populated volcanic area in the World.

## 2. The state of the art in the eruption forecast and alerts

Eruption forecast for timely evacuation is, today, the only way to defend populations exposed to volcanic risk. In fact, if people can be made safe from earthquakes provided the edifices are robust enough (which can be a difficult task only in areas of extreme magnitude events occurring very close to urbanized areas) there in not, at present, any possible defence from the most hazardous eruption products (i.e. pyroclastic flows, very fast lava flows or, in some cases, huge pyroclastic falls) other than timely evacuation, before the climax of the eruption.

Eruption forecast is often considered in principle feasible, as opposed to earthquake forecast, today considered impossible. Such statements are equally wrong, since eruption forecast is just easier to imagine, because volcanoes are well identified and localized objects easy to monitor, whereas active faults involve very large areas, and specific active patches are not evident. Actually, however, eruption forecast is still a largely empirical matter, with very uncertain outcome (Winson et al., 2014). Although some studies are starting to constrain, with physical considerations and modelling, the volcano behaviour forecast (Kilburn, 2012; Robertson and Kilburn, 2016; Kilburn et al., 2017), this is still forefront research, not easily generalizable for a practical use. The basic problem of eruption forecast, which can be dramatic in extreme risk areas (i.e. very densely populated volcanic areas) is schematically shown in fig.1. As it can be inferred from the figure, a timely forecast of an eruption occurs when a well defined alarm is given, followed by an eruption. The alarm can be considered effective, if it allows the population to be evacuated before an eruption occurs. If an alarm is issued, not followed by an eruption, it should be regarded as a false alarm. As a consequence, in contrast with an appropriate alarm (implying evacuation), two types of failure should be considered: false alarm and missed alarm. In the most general case, depicted by fig.1, a certain trend of precursory phenomena (which usually involve increase of seismicity, ground uplift and changes of chemical species and/or equilibrium in waters and in gas emissions) increases (more or less monotonically) until the eruption occurs. If the alarm for evacuation is given in the initial phase of precursors increase, there is a low probability of missed alarm, but a high probability of false alarm. In contrast, when waiting for the precursory phenomena to become very high, there is a low probability of false alarm, but a high probability of missed alarm; because the eruption could start before having the time to complete evacuation. Thus, giving the complete trade-off between false alarm and missed alarm probability, the time of alarm should be given in such a way to minimize the 'cost': which, in case of false alarm is the economic and social cost of moving away the population, whereas in case of missed alarm is the loss of human lives likely killed by the eruption. It is then very clear that, in the case of large population exposed, the probability of missed alarm can be unsustainable, for the human losses, also for very low values of eruption probability; but, in contrast, a false alarm could have an economic and social cost anyway unsustainable. Failure to correctly manage this problem, in case of high exposed value, can also imply heavy judicial responsibility for decision-maker.

The serious complexity of volcanic risk mitigation, when it is extremely high, can be further clarified by considering the low degree of confidence of actual forecasting techniques. Considering that the catastrophic Mt. St. Helens eruption can be

considered the starting point of the modern awareness of the importance of eruption forecast, a look at the most renowned eruptions in literature, occurred since 1980 can give a first insight into the problem. A good comprehensive review on the outcomes of eruption forecast as applied to volcanic eruptions since 1980 has been given in Consensus Study Report (2017). The outcomes from that report, indicate much less than 50% of success in the forecast for the most important eruptions of the last four decades. However, even in case of 'successful' forecasts, the time of alarm before the eruption should be taken into account: in fact, the most successful forecast is considered the 2000 Hekla eruption, for which a very precise timing of the eruption was predicted, but only half an hour before the eruption occurred (Stefansson, 2011). Obviously, forecasts achieved very shortly before the eruption occurrence are not of any practical use for evacuation of large numbers of people exposed.

A very complete assessment of the state of the art in the practical use of eruption forecast, evaluated through the timely issue of proper volcanic alerts, has been given by Winson et al. (2014). They analysed 194 eruptions occurred n the period 1990-2013, from 60 volcanoes, measuring the appropriateness of the issued volcanic alerts. Fig.2 shows the main results of their study: only appropriate Volcano Alert Levels issued by authorities anticipated 19% of the eruptions. Such a discouraging outcome is only a little bit higher (30%) for larger eruptions (VEI>3), increases for better monitored as well as for closed vent volcanoes, but in any case never reaches 50%. In addition, there is an average 33% of issued alerts for unrests which gave not rise to eruptions: i.e., 'false alarms'.

These results clearly show that, presently, the state of the art for eruption forecast implies it is much more probable to miss an eruption, or to give a false alarm, rather than correctly forecast it. In such conditions, the 'forecast dilemma' depicted by fig.1 becomes very dramatic to solve, in extremely risky (i.e. extremely populated) volcanic areas.

Before discussing the implications of such an ambiguous outcome for the extremely populated Neapolitan volcanoes, le's just recall the main elements of the actual emergency plans for these volcanoes, which are based on rather common procedures: 1) defining a 'red zone', which should be evacuated before the eruption starts; 2) defining a 'traffic light' system, in our case made of four colours (green, yellow, orange, red) such that the green level is the 'completely quiescent' one, and the hazard level for a possible eruption increases until, with red alert, the complete evacuation of the red zone must be realized in 72 hours (three days).

## 3. The Campanian Volcanic Zone and the Neapolitan volcanoes

The Campanian margin displays the typical features of a continental crust and lithosphere extensional domain: several normal faults, very shallow Moho (Ferrucci et al., 1989; ), high heat flow values (Della Vedova et al., 2001) and large volume ignimbrite eruptions (Di Girolamo 1968, Rolandi et al 2003; Rolandi et al., 2019a). All these features indicate the Campanian Plain forms an elongated sedimentary - volcanic plateau, 50 km long by 20 km wide, limited to the north, east and south by the Apennine chain. In accordance with these special volcanic and tectonic features, the Campanian Plain (see fig.3) has been indicated as **Campanian Volcanic Zone (CVZ)** (Rolandi et al 2003, Bellucci et al, 2006). In the last 600 ka, the CVZ has been affected by uplift and intense volcanism, alternating with periods of subsidence and marine sedimentation (Cinque et al. 1990; Scandone et al., 1991). Presently, the Campanian Plain hosts at least three active volcanic areas: Mt. Vesuvius, Campi Flegrei caldera, Ischia island. Besides such well-identified volcanic edifices, at least six ignimbrites were emplaced over the CVZ, out of the three mentioned, known volcanoes, in the last 300 ka (Rolandi et al. 2003; Rolandi et al., 2019a). The existence of a unique, large volcanic area involving the whole Campanian Plain, in which the three most known volcanoes are the ones which erupted in recent times, is supported by seismic tomography studies (Zollo et al., 1996; Zollo et al., 2008) which identified a large, thin melt layer spread at least beneath the whole Vesuvius-Campi Flegrei area, at a depth ranging between 8-10 km. The CVZ is delimited, in the Northern part, by the Roccamonfina volcano, and in the Southern part by the Somma-Vesuvius, Campi Flegrei and, offshore, Ischia and Procida islands (Rolandi et al., 2019a). The active volcanoes of the Naples district (Somma-Vesuvius, Campi Flegrei, Ischia and Procida) will be described in more

detail later. We shortly describe here Roccamonfina volcano, which is considered extinct. It is a large stratovolcano with numerous post-caldera cones within the summit caldera and on the flanks. The eruptive history consists of three main phases. The first stage built the main cone via lava flows and pyroclastics. The second stage is defined by large explosive eruptions, brown leucitic tuff, white trachitic tuffs, and caldera collapse. The third stage formed the lava domes, and scoria cones that exist on the volcanic edifice today (Giannetti and Luhr, 1983). Rouchon et al. (2008) dated summit caldera lava

domes at 170-150,000 years ago. Peccerillo (2005) gives an age range of 0.58-0.1 Ma for the whole Roccamonfina activity. Eruptive activity of Roccamonfina volcano in the last 300,000 years is then coheval to five of the six Ignimbritic eruptions generated from the CVZ.

The offshore volcanism does not involve only Ischia and Procida islands; the Campi Flegrei caldera is below the sea level for more than 50% (Somma et al., 2015), and the Gulf of Naples hosts several submerged volcanic centres (Passaro et al.,

2015; 2016). The Southernmost area then contains the three volcanoes surely active today; about three million people live to within less than 20 km from a possible eruptive vent.

## 3.1 Mt. Vesuvius

The Somma-Vesuvius volcanic complex (SV) is the most known volcano of the Neapolitan area, and one of the most famous in the World, mainly because of its ancient eruption of 79 A.C., well described by Plinius the Young (Scandone et al., 2019).

SV is a renowned volcano also because it frequently erupted in the last centuries, so becoming popular as a natural attraction in Europe, and one of the classical stops of the XIX century Italian 'Grand Tour' (Astarita, 2013). The eruptive history of SV, shown in Fig.4, is characterized by periods of frequent activity (open conduit activity) alternated with period of quiescence (closed conduit periods). The history of the SV began 0.3–0.5 million years ago. Periods of closed conduit have lasted, in the past, up to 1000 years; they are generally interrupted by plinian and subplinian explosive eruptions, which only

differ in the volume of emitted magma and the energy of the eruption. SV is composed of a multistage and older summit caldera (Mt. Somma, 1132 m a.s.l.) and a nested younger cone (Mt. Vesuvius, 1281 m a.s.l.) (Fig.3). In the last 25 ka, five plinian caldera-forming eruptions (25 ka Pomici di Codola, 18 ka Pomici di Base, 9.7 ka Mercato, 4.3 ka Avellino, and AD 79 Pompei) and at least three major subplinian eruptions (17.6 ka Pomici Verdoline, AD 472 Pollena, and AD 1631) occurred (Rolandi, 1998). The last cycle of open conduit activity started in 1631 and lasted until 1944. Since 1944, this area

has experienced a very large urban development: towns around the volcano grew of about three times in population, becoming seamlessly connected to the city of Naples. It is important to note that towns around Vesuvius, which were primarily tourism-oriented before the II World War, are now very busy outskirts of Naples hosting resident population. The present activity of this volcano, otherwise quiescent, only consists of a background seismicity seldom overcoming magnitude 2, with maximum magnitude M=3.6 reached by an earthquake occurred in October 9th 1999, during a period of increased

seismicity rate and magnitude, lasted some months (between 1999 and 2000; see De Natale et al., 2004). A convincing explanation for background seismicity, at this and similar composite volcanoes, was given by De Natale et al. (2000); it is interpreted as due to the gravitational stress, due to the volcano topography load, which is focused along the sharp rigidity contrast between the external rocks, made of explosive pyroclastic deposits, and the central conduit made of solidified magma. At Vesuvius however, except for the increased seismicity occurred in the mentioned period, no other signals of

anomalous activity have been ever recorded at this volcano (i.e. ground uplift, geochemical anomalies, consistent LP seismicity, seismic tremor, etc.).

The volcanic hazard at SV has been thoroughly described, in a probabilistic framework: firstly by Rossano et al. (1998) and then, in a more complete way, by De Natale et al. (2005). The main volcanic hazards are pyroclastic flows and ash/pumice fallout (see also Sacchi et al., 2019; 2020), but also associated hazards like earthquakes, lahars, lava flows and floods, need

to be considered (Sacchi et al., 2009). In particular, large floods caused by the re-mobilization, due to heavy rains, of the old,

loose pyroclastic deposits on the topographic reliefs around Vesuvius, caused almost total destruction and 160 casualties in May 1998 (Mazzarella and Tranfaglia, 2002).

Somma-Vesuvius volcano was the first one for which a complex emergency plan had been issued, by Italian Civil Protection, in 1995. The features of this first plan were almost the same of the present ones (Dipartimento Protezione Civile, 2015), although some minor modifications have been made in the last decades. The eruptive scenario, used to define the red zone (shown in fig.5), was a sub-plinian eruption, mainly because, at the time in which the first plan was released, the idea that a shallow magma chamber was almost constantly fed by deeper magma was predominant in literature (e.g. Santacroce, 1983), and related computations of the volume of magma fed from 1944 were consistent with a sub-plinian eruption. Later on, such a volcanological hypothesis has been heavily questioned, mainly as a consequence of seismic tomography studies which failed to identify such a shallow chamber filled with molten magma (Zollo et al., 1996). In particular, Marzocchi et al. (2004) pointed out that, from probabilistic estimates, after 60-200 years of quiescence a violent Strombolian eruption (VEI=3) is the most probable event, whereas a subplinian one (VEI=4) has a probability to occur much lower and a Plinian one (VEI=5) has still a probability >1%. However, in the emergency plan the sub-Plinian scenario has been maintained. The updated red zone of Somma-Vesuvius hosts today about 700.000 resident people, and totally or partially includes 25 municipalities (see Italian Civil Protection website: http://www.protezionecivile.gov.it/media-communication/dossier/detail/-/asset_publisher/default/content/aggiornamento-del-piano-nazionale-di-emergenza-per-il-vesuvio).

### 3.2 Campi Flegrei Caldera

Campi Flegrei caldera, located in the Southwestern part of the Campanian Volcanic Zone, contains the Western District of the city of Naples (see fig.3). It is a collapse caldera, formed by the huge eruption of Neapolitan Yellow Tuff, occurred about 15.000 years BP (Rolandi et al., 2003; 2019a; De Natale et al., 2016). Neapolitan Yellow Tuff, a VEI=6 event with an erupted volume of about 40 km$^3$, generated pyroclastic flows which represent the main eruptive products found in the Naples province, that have been the main building material in this area for more than 2.000 years, before the concrete became diffused. One of its facies, named 'Pozzolana' (from the name of the town, Pozzuoli, located at the caldera center) has been the main element composing the famous 'Roman cement' which allowed ancient buildings of the Roman age to be so resistant and long lived (Vanorio and Kanitpanyacharoen, 2015). Campi Flegrei caldera has been long thought to be firstly formed by the larger European eruption ever known, namely the Campanian Ignimbrite, a VEI=7 event whose erupted volume estimates range between 150-300 km$^3$ (Rosi and Sbrana, 1987; Orsi et al., 1996). However, Rolandi et al. (2019b) recently demonstrated that Campanian Ignimbrite main products (Grey Tuffs) were erupted from the Campanian Plain, North of Campi Flegrei, and did not cause any caldera collapse. Campi Flegrei caldera eruptive activity has been dominated by phreato-magmatic eruptions, whose explosivity is due to the contact of rising magma with the large geothermal system located beneath the caldera, down to about 2.5-3.0 km (Rosi and Sbrana, 1987; Piochi et al., 2014). The collapsed area, as recognized by geophysical data (Cassano and La Torre, 1987; Capuano and Achauer, 2003) has a radius of about 3 km, with center approximately located at the Pozzuoli town harbour; more than 50% of the caldera is below the sea level (Somma et al., 2015; Steinman et al., 2018; Sacchi et al., 2019). All the eruptions younger than 10.000 years are located within such an area, more frequently occurring from its borders, marked by buried caldera ring faults (De Natale and Pingue, 1993; De Natale et al., 1997). The eruptive history of Campi Flegrei is sketched in fig.6. The only eruption in historical times occurred in 1538, and this is the reason why this volcanic area is by far less renowned than Vesuvius. However, this area is also characterized by unrest episodes with large uplift and subsidence of the ground level. In the last 2.000 years, subsidence, at an almost constant rate between 1.5 and 2.0 cm/year, has been generally dominant, except during about one century preceding the 1538 eruption, when fast uplift occurred, at an average rate about one order of magnitude larger than the secular subsidence (see Fig.7). Another episode of fast uplift, but without a subsequent eruption, has been inferred by Morhange et al. (2006) between the VII and the VIII century. Starting since the 50's of the last century, the ground

movements in the area again reversed to uplift, which, since 1950 to 1984 totalled about 4.5 m with peak rates, in 1983–1984, of about 1 m/year. After about 20 years of relatively fast subsidence following the 1984 peak of vertical ground displacement, uplift started again around year 2005 (Fig.7), at rates comparable to those of subsidence (on average 9 cm/year), but much lower than previous uplifts (Moretti et al., 2018; Troise et al., 2019). Both the post-1984 subsidence and the subsequent and still ongoing uplift phase, showed minor, short-lived peaks of uplift followed by a fast recovery of the whole uplift (the so-called 'mini-uplift' episodes: see Gaeta et al., 2003; Troise et al., 2007; Iuliano et al., 2015). The intermittent uplift phases, started in 1950 and still ongoing, show a cumulative uplift, in about 70 years, consistent with an average rate of 0.075 m/year. The average uplift rate computed from the total uplift observed since about one century and half before the 1538 eruption, although in the very rough approximation of secular inferences, is about 0.1 m/year, which is the same order of magnitude and not clearly distinguishable considering the large uncertainties. The 1950-2019 unrest, with intermittent ground movements and seismicity, gives a very precise idea about the large uncertainty to identify true eruption precursors in this area (Moretti et al., 2013; Troise et al., 2019; Moretti et al., 2020). In 1970, during the first, fast uplift episode clearly identified, the urban area of Pozzuoli closest to the harbour, namely the 'Rione Terra', was completely evacuated, and never inhabited anymore. In 1984, the whole town of Pozzuoli was evacuated (in the newly built town of Monteruscello), and spontaneously re-occupied some months after when seismicity stopped and ground started to subside (Barberi and Carapezza, 1996). Today, after about 15 years of uplift at an average rate of about 0.08-0.09 m/year, and after about 60 years of intermittent unrest, it is not clear yet if Campi Flegrei caldera is going towards an eruption (Chiodini et al., 2016; Kilburn et al., 2017; Forni et al., 2018; Troise et al., 2019). Campi Flegrei then represents a typical example of an area where the short-term forecast is made particularly difficult due to the problem of distinguish between hydrothermal and magmatic effects during unrest (Moretti et al., 2017; 2018; 2020). Troise et al. (2019) recently demonstrated that the 1982-1984 unrest was accompanied by a shallow magma intrusion (with volume of about 0.06 km$^3$), whereas the present unrest (started in 2005) cannot be ascribed to shallow magma migration. It is due to the rising of magmatic gases and consequent heating of the shallow aquifers. Whatever the physical explanation of the on-going unrest, however, the area is in the alert level 'Yellow' ('Attention' level) since December 2012. Recent unrest events, however, naturally require a well organized response at least in the average-long term, because we cannot actually predict the possible, future evolution of the state of the volcano.

It is worthy to note that the emergency plan for Campi Flegrei, actually on the way to be completed, at the end of 2012 did not exist yet, not even the red zone was defined (it was officially released in 2015, see Department of Civil Protection website). This probably occurred because, differently from the first edition of Vesuvius emergency plan of 1995, there was no idea of a given eruptive scenario for this area, not even the eruptive vent could be defined because it could be everywhere in the caldera area. In fact, Rossano et al. (2004) firstly suggested to use a probabilistic scenario made up of any possible kind of eruption from any possible vent spanning the caldera area; the probability of each eruption type was inferred from its frequency in the last 10.000 years. Rossano et al. (2004), using a rigorous Bayesian approach and a simplified modelling technique for pyroclastic flows on the actual topography, first obtained a probability hazard map which clearly indicated, as the most probable zones experiencing pyroclastic flows, an area very similar to the presently defined red zone. More accurate results, using the same methodology, were further obtained by Mastrolorenzo et al. (2006b). However, their hazard maps were never considered by Civil Protection authorities, at the time, and only 11 years after they officialised the red zone, based on a paper by Neri et al. (2015), which used a very similar probabilistic approach (although with a much more approximated and less rigorous pyroclastic flow modelling technique). The only difference between the two methods (besides the oversimplification of pyroclastic flow modelling) is that Neri et al. (2015) assumed a non uniform probability for vent opening, based on the assumptions and results of Bevilacqua et al. (2015). The final results are, regarding the definition of the red zone, very similar to the results obtained by Rossano et al. (2004) and Mastrolorenzo et al. (2006b). The red zone of Campi Flegrei hosts today about 600.000 resident people, and totally or partially includes 6 towns and several

suburbs of Naples city (see Italian Civil Protection website: http://www.protezionecivile.gov.it/attivita-rischi/rischio-vulcanico/vulcani-italia/flegrei/piano-nazionale-di-protezione-civile).

## 3.3 Ischia island

Ischia island, located South-West of Campi Flegrei, is another volcanic field, characterized by both effusive and explosive eruptions (see Passaro et al., 2015). Eruptions here, in fact, range from lava flows to phreato-magmatic ones, thus being half-way between the Vesuvius and Campi Flegrei eruption styles. The Ischia volcanism developed between about 130–150 ka BP (Vezzoli, 1988) and 1302 AD (de Vita et al., 2006; 2010). Fig.8 shows the eruptive history of Ischia island. The volcanism in the island is strictly linked with the resurgence phenomena of the Mt. Epomeo, an horst which is thought to move up and down (with dominant uplift in the past, due to its prominently high topography). Resurgence has been ascribed to repeated injections of magma at depths of 2–3 km, where a laccolite magma chamber is hypothesized (Orsi et al., 1991; Cubellis and Luongo, 1998; Tibaldi and Vezzoli, 1998; Acocella and Funiciello, 1999; Molin et al., 2003; Carlino et al., 2006; Paoletti et al., 2009; Sbrana et al., 2009;). The last eruption occurred in 1302, so common people often does not recognize Ischia as a volcano. The movements of the Epomeo horst also cause slip on the bordering faults, which decouple it from the rest of island. For this reason, Ischia has been struck in the past by several catastrophic earthquakes that, although not so high in magnitude, are very shallow and then very destroying within short distances (see Table 1). The most destructive earthquake at our knowledge has been the July 23$^{rd}$, 1883 event, which completely destroyed the town of Casamicciola and parts of neighbouring towns, killing 2313 people (De Natale et al., 2019). This catastrophic event was preceded, two years before, by a slightly smaller event, killing anyway 126 people; these two larger events were the final ones of a sequence of 6 large earthquakes started in 1828 (which killed 56 people, see De Natale et al., 2019). After 134 year of negligible microseismicity, a new larger earthquake occurred on August 21$^{st}$, 2017 (De Natale et al., 2019) killing two people and causing severe damages in a specific area between Casamicciola and Lacco Ameno (the same one experiencing the most severe damages and/or destruction from the major earthquakes of the XIX century). De Natale et al. (2019) also warned that the 2017 earthquake could represent the beginning of a new major earthquake sequence, just like the XIX century one, further suggesting detailed and urgent mitigation measures. The occurrence of the 2017 earthquake also raises the question if such major seismicity could be linked to magmatic pressure increase. Looking at the past activity, it does'nt appear any clear correlation among major earthquakes and eruptions, but we don't know much about the eruption precursors, since the last eruption dates back 1302. Actually, there is not even agreement about the dip of the fault plane and earthquake mechanism (Nappi et al., 2018; De Novellis et al., 2018; Calderoni et al., 2019), which is critical to understand if the event corresponds to an uplift pulse of Epomeo horst (which could reflect a pulse of increased magma pressure) or to cumulated subsidence (in the last decades Epomeo horst is subsiding, at an average rate on the order of 0.1-0.5 cm/year).

Despite the high hazard posed by seismic and volcanic activity in the island, which hosts about 70.000 resident people in only 46 km$^2$, the risk here has been generally understated, and in fact there is no emergency plan for volcanic eruptions, nor an appropriate plan to secure urban areas from seismic risk has been undertaken till now (besides specific suggestions and warnings given by De Natale et al., 2019). Ischia island also represents the absolute needing for planning an eventual evacuation by sea routes, which have been till now, in the Vesuvius and Campi Flegrei emergency plans, completely neglected.

## 4. The Emergency Plans: short history and description

Emergency plans for the Neapolitan volcanic areas date back 1995, the year in which the first plan, related to Mt. Vesuvius, was released. The scenario used for a next eruption of this volcano was a sub-plinian one, i.e., taking as reference the large eruption of 1631, which opened, after several centuries, the eruptive period ended with the last eruption of 1944. At the time when the plan was elaborated, the prevailing scientific opinion was that the main magma chamber were refilled at a constant

rate (Santacroce, 1983); based on such assumption, the computed magma volume after 50 years gave almost the same value than the erupted volume computed for 1631. Based on such a scenario, the red zone was defined, as the one which could be hit by pyroclastic flows during a sub-plinian eruption (similar to the 1631 one). In more recent times, such a model of constant magma chamber replenishment has been abandoned, although the red zone has remained almost the same. Besides the red zone, the most prone to pyroclastic flow hazard, a yellow zone has been defined, as the most prone to large thickness of pyroclastic falls (i.e. ash and pumice). Since the first emergency plan of 1995, some basic assumptions have been established and used for subsequent plans, for instance the alert levels. Actually, there are four levels of alert: Green (basic level, no anomalies), Yellow (first alert, some anomalies), Orange (pre-alarm, several anomalies), Red (alarm, evacuation). The red zone of Mt. Vesuvius at the time contained about 800,000 people. The definition of the red zone for Vesuvius gave rise, since it was released, to strong (and sometime harsh) scientific and political discussions; however, in literature, the scientific debate about the emergency plan, and more in general about the volcanic hazard and risk in this area, can be synthesized by few papers (e.g. Scandone, 1993; Rolandi, 2010; Papale, 2017;. The main objections to that plan questioned the choice of the scenario and the possibility to really evacuate 800,000 people in three days, as claimed on the plan. Most of the criticism addressed to the choice of a single scenario, based on a sub-plinian eruption, indicated by a constant magma feeding model that was not supported anymore in literature since the years 2000. Alternative scenarios, based on probabilistic (Bayesian) hazard estimation from the whole spectrum of possible eruptions, were firstly proposed by Rossano et al. (1998), but never adopted. The present red zone is shown in fig.5 and contains about 700,000 people. Since 2015, also the red zone for Campi Flegrei eruptions has been released (fig.5). Differently from the Mt. Vesuvius one, the red zone for Campi Flegrei has not been designed on the basis of a specific scenario, but taking into account, in a probabilistic approach, the whole spectrum of possible eruptions (Neri et al., 2015); just like Rossano et al. (1998) had firstly suggested for Vesuvius, and then for Campi Flegrei (Rossano et al., 2004). The Rossano et al. (2004) probabilistic approach for Campi Flegrei pyroclastic flow hazard map, developed eleven years before the study adopted to define the red zone (Neri et al., 2015), gave very similar results for the area with maximum probability to be hit by pyroclastic flows, and then for the red zone.

What is also relevant, for the subsequent considerations, is that both the two existing Emergency Plans (for Vesuvius and Campi Flegrei), regarding the evacuation plan in case of red alarm, relies only on the use of land transportations, not considering the use of ships. This choice, probably driven by the concern for possible tsunamis accompanying eruptions (anyway extremely rare in such areas, and surely not expected before the eruption), cannot be equally applied to an Emergency plan of the Ischia Island that, however, is the only remaining volcanic area where there is not yet any Plan.

Natural hazards at Ischia Island have always been surprisingly underestimated. Despite the catastrophic earthquakes occurred in the past in the Casamicciola area, including the devastating one of 1883 causing about 2300 casualties (De Natale et al., 2019), there were no monitoring seismic stations on the island until 1993, when the first permanent monitoring network was installed in three sites, then improved with the fourth site in 2015 (De Natale et al., 2019). The recent earthquake occurred in August 2017, as already discussed, poses new heavy concern, both regarding the vulnerability of urban areas (De Natale et al., 2019) and because, at present, the relation of such seismicity with eventual volcanic processes is not clear at all (Calderoni et al., 2019; Nappi et al., 2018; De Novellis et al., 2018).

Coming back to describe more details of the Emergency Plans, it is important to note that the first three steps of the alert: Green, Yellow, Orange, are decided by the National Civil Protection, normally upon advice from the National Commission for High Risks, whereas the last step, from Orange to Red (implying complete evacuation of the red zone in 72 hours) is only decided by the Italian Premier (Dipartimento Protezione Civile, 2015). Regarding the evacuation plan, which starts once the Red alert is declared, evacuated people is meant to be distributed in all the Italian Regions, according to a correspondence between each municipality and a given Italian Region. There is however no other detail in the Plan, programming, for instance, exactly where (i.e. in which houses, hotels, etc.) people will be relocated in each Region.

## 5. Evacuation Plans: strength and weakeness

We do not aim to discuss in detail the Emergency Plans in their intermediate steps. We only focus on what should occur after the declaration of the Red alert, which implies the rapid (within 72 hours), complete evacuation of the red zone. As already mentioned, the red zones of Campi Flegrei and Vesuvius presently contain, respectively, about 600,000 and about 700,000 people. The evacuation plans, in the present formulation, state they must be evacuated by roads, on-land. Most of residents in the red zones (and also out of them) are sceptic about the real possibility to successfully evacuate such a high number of people within three days (Solana e al. 2008;Carlino et al., 2008). They believe it would be not possible, both because of the likely massive panic that would spread across, and for the huge traffic which characterizes the few main roads to evacuate, even in normal days. In the turmoil that would likely accompany a massive evacuation, it is easy to imagine those roads completely jammed. These are, however, just feelings of the people, and we will assume here the evacuation can be successfully organized. There are anyway two former, successful example of evacuation in the Neapolitan area, both of them in the Campi Flegrei area. The first evacuation occurred in 1970, at the beginning of the first recent (recognized) large Campi Flegrei unrest of the period 1969-1972. About 3000 people were forcedly evacuated, in just one day, from the Rione Terra, a district of Pozzuoli just behind the Port of Pozzuoli, which was at the time (and also in the following unrest episodes) the area of maximum uplift. After that episode, in 1984, when the subsequent unrest was rapidly progressed and continuous earthquakes caused extreme concern, the whole town of Pozzuoli, about 40,000 people, was evacuated and transferred in a new town: Monteruscello, located about 3 km NW and built in few months to host the Pozzuoli citizens (Barberi and Carapezza, 1996). The evacuation of Pozzuoli town was, in the opinion of several experts, probably the most successful operation of Civil Protection in Italy. However, it involved more than an order of magnitude less people, with respect to the present red zone of Campi Flegrei (or, equivalently, of Vesuvius). In addition, the main productive activities (factories) in the Pozzuoli area were not stopped, and evacuated people could go to work anyway in the 'red zone' of that time. Finally, while the Rione Terra was never let to be populated again, the complete evacuation of Pozzuoli lasted about one year or less, and after that period the town was fully populated again, because people almost spontaneously came back. The red zones defined in the present emergency plans are chosen in order to take into account the largest eruptions having non negligible probability, so involving very large numbers of people; and some volcanologists ask for even higher precaution (e.g. Mastrolorenzo et al., 2017). Actually, however, defining very large red zones to be evacuated before an impending eruption, could seem a better caution, but it also makes the evacuation decision a much heavier responsibility to assume; dramatically costly in case of false alarm. The success of the first two evacuations was undoubtedly due to the limited area, and limited number of people involved. Each of them was in effect a single step following the concept of a progressive evacuation; in which areas progressively larger could be involved, following the eventual increase of anomalous phenomena and/or the starting of phreatic activity or magmatic eruption. The concept of progressive evacuation has been actually implicit also in the few successful evacuations before eruptions, which proceeded in steps, enlarging the evacuated areas in response to the increase of eruptive activity (Tayang et al., 1996).

Another choice which could be surely debated, about the effectiveness of the present evacuation plan, is the lack of evacuation by sea, with large ships which could rapidly move a lot of people without any traffic problem. This choice is probably due to the fact that the ports in Campi Flegrei and Vesuvius towns (except for the port of Naples) are not suited to host large ships; another obstacle is probably thought to be the possibility that tsunamis may accompany the eruptions. The evacuation by the sea, anyway, is the only one that can work for an evacuation plan of Ischia Island, which is compelling and, sooner or later, must be done. Moreover, looking at historical eruptions of Neapolitan volcanoes, it turns out there is no evidence for any tsunamis associated to their eruptions.

We proceed now to make clearly evident the main problems of the present day Emergency Plans. As already said, we will not discuss the steps from Green to Orange, nor we want to assess the details of the first evacuation phase, i.e. the way to move 600,000-700,000 people out of the red zone. Regarding the present choice to move people exclusively on-land, we just

noted that evacuation by sea, using large cruise ships, would be much more rapid and effective, avoiding the multiple problems linked to the traffic and to the lack of appropriate roads.

We want, instead, to discuss here two problems, which are also in some way interrelated: the extremely high number of people to evacuate in case of an impending eruption, and the lack of plans, today, to reallocate such a high number of evacuated people taking into account realistic times people will have to spend out of their homes in the red zone.

Regarding the first problem, namely the high number of people to evacuate, it is clear that the decision-makers have to take a very big responsibility to declare the Red alert, which will cause dramatic social problems and economic damages. The

economic loss per each year the evacuation lasts can be reasonably estimated by considering that 600000 people are almost 1% of the total Italian population. So, by suddenly stopping the economy produced by 600000 people would represent a loss of 1% of the Italian PIL. Since the annual Italian PIL is around 2000 G€, 1% is about 20 G€. To such high cost it should be added the cost of assistance to the evacuated people (i.e. travel, hosting, subsistence, services, etc.), whose minimum estimate (15-20 k€ per year per person) gives another 10-20 G€/year. A total cost in the range 30-40 G€/year (for Campi

Flegrei; for Vesuvius it would be about 20% larger) represents the amount of one of largest annual financial package of Italian Government; so, it is likely unsustainable, even for just one-two years. But the real problem, fundamental also to evaluate the real amount of social disease and total economical loss, is the second one: how much time will such a large number of people spend out of the original towns? To answer this question, we can consider two possible cases:

1) the eruption occurs in short times after the alert

2) the eruption does not occur in short times

In the first case, it is clear that a considerable part of the evacuated area will be destroyed or anyway seriously affected, so that several years, probably decades, will be needed to restore conditions to make liveable again the area. But, anyway, the occurrence of an eruption would likely indicate a new state of the volcano dynamics, making even more unpredictable its subsequent activity. A clear example of such a long lasting eruptive phase, for a volcano which was quiescent since 400

years, is the case of Soufriere Hills in Monserrat, erupted for the first time in 1995, evacuated since then and still in alarm because experiencing consecutive eruptions (Smithsonian Institution website:

https://volcano.si.edu/volcano.cfm?vn=360050 and references therein)

Regarding Neapolitan volcanoes, volcanologists normally assume that, after a long non-eruptive period, a new eruption of Mt. Vesuvius will open a new cycle of eruptive activity, which can become much more frequent like it was during the XVII

to XX centuries period (Santacroce, 1983). For Campi Flegrei and Ischia, which are dormant since several centuries, the occurrence of an eruption today would make equally much more unpredictable the future evolution of volcanic activity. All these considerations make very evident that, in the event 1, realistic times to repopulate the red zone would be extremely long or indefinite, on the basis of objective considerations.

What would happen if, on the contrary, the eruption would not occur in short times after the evacuation (event 2)? In this

case, we have to consider that, if the precursory signals were so strong and evident to convince decision-makers to evacuate 600,000-700,000 people causing a real 'disaster', in an economical and social sense, they could certainly not decide to put again people at the same high risk situation, considering that times of preparation of an eruption, are mostly unknown but can certainly be very long in some cases. Also in this case, a typical example of what could happen is given by the unrest episodes at Campi Flegrei. We know that intense uplift episodes started already in 1950, although apparently that unrest was

not noted. However, after the unrest of 1969-1972, people thought the danger were over, even if the 'Rione Terra', the urban area very close to the Port, which was evacuated at that time, was never re-populated again. After about 10 years, a new unrest episode started, with even higher rates of uplift and much more intense seismicity (De Natale et al., 1991). At the end of 1985, people again thought the danger was over, and Pozzuoli town was populated again after the evacuation. However, after about 20 years, a new unrest episode has started, still on going today, which again poses large concern in the

volcanologists, authorities and population. In practice, the alert at Campi Flegrei lasted 70 years till now, and is still on-

going; the first evacuated zone, Rione Terra (3000 people evacuated), has not been repopulated; Pozzuoli was, but it must be noted that only 40,000 people were evacuated, that the main economic activities were not stopped, and that Monteruscello (the new town hosting evacuated people) was very close to Pozzuoli and 'inside' the Pozzuoli municipality. In case of 600,000-700,000 people, scattered all along Italy (as the present evacuation plan prescribes), it would be impossible, today,

to consider they could come back to their homes in similar conditions.

Finally, let's make a comparison with another, catastrophic emergency recently occurred because of COVID-19: it caused an economic damage, for Italy, on the order of some hundreds millions of euros in a lockdown lasted 2-3 months. With a not programmed, sudden massive evacuation of Campi Flegrei or Vesuvius, as presently planned, the same amount of economic loss would be totalled in the first few years. However, such a loss would be, in this case, accompanied by an additional,

catastrophic social unrest due to the uprooting and provisional reallocation of almost a million people.

## 6. Elements for a reliable Evacuation Plan and Emergency Management

The nature and size of volcanic hazard in the Neapolitan areas, as well as the experience of previous evacuation inside the Campi Flegrei area, give important suggestions on how to build a really working Emergency Plan. The previous experiences of evacuation inside the Campi Flegrei area were successful (although no eruption occurred), but limited to 3000-40000

people. Increasing the number of evacuated people by 1 to 2 orders of magnitude, although it could seem to be more conservative with respect to the possible occurrence of larger eruptions, introduces additional, very huge problems. They are related, how we explained in the previous paragraph, to the extreme responsibility taken by the decision maker, in terms of economic and social costs, as compared to the high uncertainty about the evolution of volcanic phenomena. These problems necessarily translate into very long times of permanence of evacuated people out of the red zone, in case of evacuation. Such

times can be estimated, in the most optimistic way, in the order of many years or decades. This means that the evacuation plan cannot simply provide that all the people goes safely away from the red zone: it must provide a sort of 'second life' for the evacuated people, which must live in the new place for decades, perhaps forever. Obviously, in this case it is not realistic to assume (as the present plan implicitly does) that several hundred thousands people can live for decades as refugees, in temporary accommodation like hotels, etc., assisted by the Government. Making some simple (and optimistic) calculations,

besides the unbearable social unease, the economic costs of such a condition would be on the order of 30-40 billions euros per year. The amount of economic and social costs of an evacuation from one of the two main volcanic areas, operated as imagined till now, thus clearly demonstrate this is not only a problem for Italy, but surely of European scale.

It should be now clear that the problem of volcanic hazard in the Neapolitan area cannot be afforded in the way it has been thought till now. In view of a rational approach to this incredibly hard problem, some basic conditions should be reached

well before the starting of a volcanic crisis possibly leading to an eruption. The most basilar conditions are:

1) the number of residents in the red zones must be decreased;

2) the urban areas in the red zones must be made less densely populated and chaotic, with large roads, escape routes and edifices resistant to seismicity which accompanies volcanic unrest;

3) the evacuation of the population must be completely organized well before the crisis: all people should be assigned

a new home, a new working perspective, and all the services for living there for many years/decades, likely forever (schools, hospitals, medical care, leisure's, etc.).

The first two points are fundamental in order to make really feasible a massive evacuation in case of red alert, and to protect the population from the most common phenomena (mainly earthquakes) occurring during unrest and pre-eruptive phases. The third point is, on the contrary, compelling to avoid that a possible evacuation would result in a social and economic

disaster. However, a careful prior organization of a future evacuation, for all the population (point 3), may also help to afford the problem at point 1, and consequently the problem at point 2. In fact, a prior organization of the 'second life' of people in case of evacuation may convince several people, if incentivized in some way, to abandon in advance the red zone, well

before any significant official alert. A significant decrease of residents in the red zone (point 1) will make easier to re-organise and re-planning the urban areas, making them more resistant and resilient (point 2). Associated to these measures,

another important improvement of the Emergency Plans would be to introduce the concept of 'progressive evacuation'. At present, only massive, total evacuation of the whole red zone is considered in the Emergency procedures. As we already discussed, deciding to move several hundreds thousands people is a very huge responsibility for decision makers; in particular because, even in presence of strong anomalies which can be considered pre-eruptive signals, the probability of false alarm is extremely high: probably anyway higher, even very close to the eruption time, than the probability of eruption.

The experience of the past, and in particular the two successful but limited evacuations in the Campi Flegrei area (1970 and 1984, respectively 3000 and 40000 people) suggests to operate a progressive evacuation, which starts in a limited area, where precursory signals (and/or prior data) indicate the highest probability of eruption and/or of phreatic explosions, and then proceeds in progressively larger areas if the pre-eruptive signals increase (or first eruption phases start/progress). Past examples of successful evacuation (i.e. Pinatubo 1991, see Tayang et al., 1996) operated in a progressive way, by enlarging

the evacuated area following the evolution of the eruptive activity. Such a procedure has the advantage to allow to evacuating the most hazardous areas without causing disastrous social and economic consequences and, in particular, without to be pushed to wait for macroscopic unrest signals (in the hope to absolutely avoid false alarms). When operating with progressive evacuation, in the first steps (with relatively few people evacuated) residents could be let free to choose if definitively abandon the red zone, proceeding to the planned 'second life', or to wait for some time in temporary housing,

likely not very far from the evacuated area.

The association of prior programmed 'second life' of evacuated people and of the progressive evacuation could hence work very well, in cases similar to the very long and variable 1950 to present Campi Flegrei unrest, to help decreasing the number of residents and to allow improving the urban resilience in the risky areas.

It should be noted anyway that moving away several hundred thousand people to prevent disasters, to reallocate them in new

permanent positions, is however a formidable goal, which can only be obtained during a well programmed, long enough period. Experiences of massive reallocations of whole urban centres are very rare, and normally they occur after disasters, to reconstruct the partially destroyed towns in new sites. We could find in literature only two examples of displacement of whole urban centers to prevent disasters. The first case is Valmeyer, Illinois,1245 inhabitants, where population, after a catastrophic flood of the Mississipi in 1993, moved the whole town at a new site in 1995 (Rozdilsky, 1996). The second case

is Kiruna, Sweden, 23,000 inhabitants, is another town which is going to be moved 3 km apart, because of hazard posed by ore activities which are causing continuing ground sinking and felt seismicity (Dineva and Boskovic, 2017). Dineva, S & Boskovic, M 2017, 'Evolution of seismicity at Kiruna Mine', in J Wesseloo (eds.), Proceedings of the Eighth International Conference on Deep and High Stress Mining, Australian Centre for Geomechanics, Perth, pp. 125-139, https://doi.org/10.36487/ACG_rep/1704_07_Dineva).

**Conclusion**

The Neapolitan volcanic area, with three explosive volcanoes and about three million people closely exposed, has the largest risk in the World. The volcanic risk is here associated with other risks, the main one being the seismic risk. Risk mitigation in this area is, for these reasons, a paradigm to manage all the situations of densely populated volcanic areas in the World. It is very clear that, given the present state of the art of volcanology, volcanic risk mitigation in densely populated areas cannot

rely only on eruption forecast, still based on empirical procedures largely uncertain and not even really quantifiable in a probabilistic way. For this reason, we suggest here that effective mitigation procedures must, in these cases, to be flexible enough to take into account economic, social and political considerations in addition to volcanological ones. In fact, in densely populated areas one is faced by the double problem of low reliability of forecast and no possibility to estimate the size of the eventual eruption. In the present emergency plans for Neapolitan volcanoes, the probability of missed alarm is

practically neglected, and the 'red zones' (i.e. the areas to be quickly evacuated before the eruption) are assumed very large on a precautionary base, in order to manage the occurrence even of the largest eruptions, unless considered very unlikely. These two assumptions are both very critical: the first one is demonstrated to be simply wrong, from all recent eruptions experience; the second one, in the light of volcanological considerations, is demonstrated to make the evacuation decision too heavy for decision makers, because potentially catastrophic in economic and social terms, mainly considering the high probability of false alarm. Once we put in evidence the clear faults of the present emergency plans, we show what should be the guidelines for making them really effective. The first, essential requirement for Neapolitan area, is to decrease population in the most exposed urban zones, well before any volcanic emergency. There are economical considerations, only mentioned here and to be deepened with the help of economists and social scientists, which could make feasible such a difficult task in this region. Another imperative action is to improve the quality of buildings, reinforcing them to be resistant to earthquakes which unavoidably precede and accompany both eruptions and unrest episodes. Once the closest urban areas are made more resistant and hence more resilient, the possible evacuation before an impending eruption must be thoroughly programmed in advance, so to minimize the economic and social impacts. A really feasible emergency plan, in addition, should consider a 'progressive' evacuation, which would start from the most risky area and then would progressively proceed to farther areas, if the precursory signals increase or eruption starts.

Besides the peculiarity of the Neapolitan area, the most striking guidelines to be followed in any densely populated volcanic area can be summarized in few important points:

if possible, decrease the resident population density in the most risky areas;

improve the resistance of the buildings and, in general, the quality of infrastructures and the resilience of urban centres;

plan in great detail the possible evacuation, well before any emergency, by taking registered all the population and its changes over time, so that it can be relocated, with jobs and services, in the new places. It will minimize economic and social costs;

identify, well before the emergency or, in large polygenetic volcanic fields, as soon as possible, the most risky areas very close to the possible vent opening. Then, rather than a massive total evacuation of the red zone, plan a progressive one, starting from the areas closest to the likely vent and involving the farthest from it only when precursory signals are extremely evident or the first eruptive phases start.

In conclusion, mitigation of extreme volcanic risk in densely populated areas require a widely multidisciplinary approach, which starts with volcanological considerations but heavily involves several disciplines: economy, social sciences, city planning, information technology. Actually, nearly 70 cities with populations exceeding 100,000 live with the threat from volcanic eruptions (Heiken, 2013); so, the elements and conclusions presented here will have value worldwide.

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

**Author Contribution**

GDN prepared the manuscript with contributions from all co-authors.

**Competing Interests**

The authors declare that they have no conflict of interest.

**Ackowledgements**

We gratefully acknowledge the constructive reviews by Roberto Moretti, Marco Sacchi and Giuseppe Rolandi, as well as the helpful comments by Chris Kilburn and Paolo Harabaglia. They significantly improved the first version of the paper.

**FIGURES**

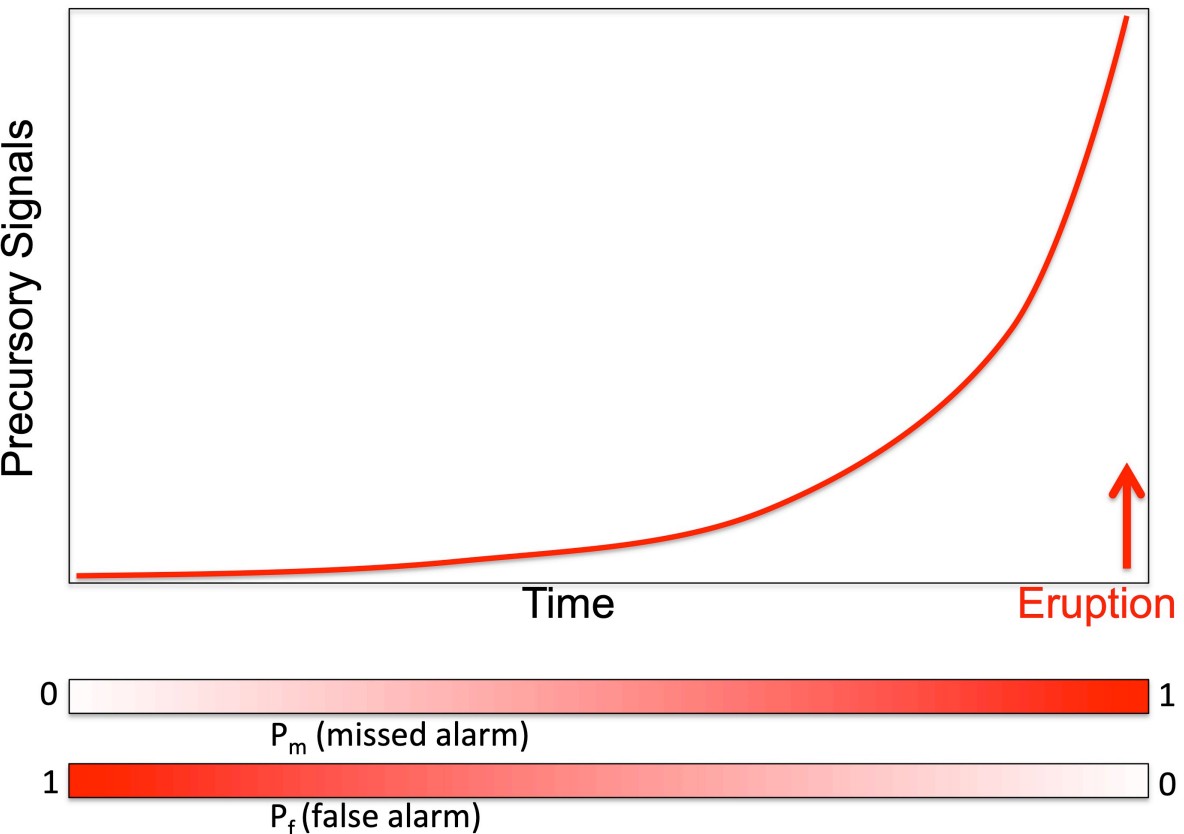

**Figure 1:** Sketch of an 'ideal' evolution in time of precursory signals before an eruption. Declaring an eruption alert shortly after the beginning of precursory signals increase, the probability of false alarm is very high, but the probability of missed alarm is low; on the contrary, declaring an alert only when precursory signals are extremely high, the probability of false alarm is low, but it is very high the probability of missed alarm (in the sense of 'too late'), because eruption could start suddenly, before any civil protection measure (normally, evacuation of most risky areas) can be completed.

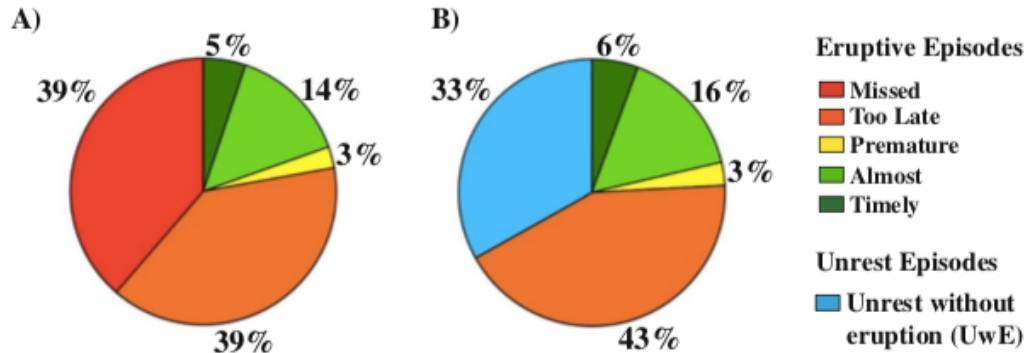

**Figure 2:** Percentages (in terms of relative frequencies) of successful and unsuccessful volcanic forecasts and warnings (from Winson et al., 2014). A) Relative frequencies of the eruption forecasts in each category, associated to a colour. Note that successful forecasts (Timely and Almost timely, in two gradations of green) are only 19% of the total. B) proportion of alerts for Unrests without Eruption (false alarms), as a percentage of all categories but the first ('Missed').

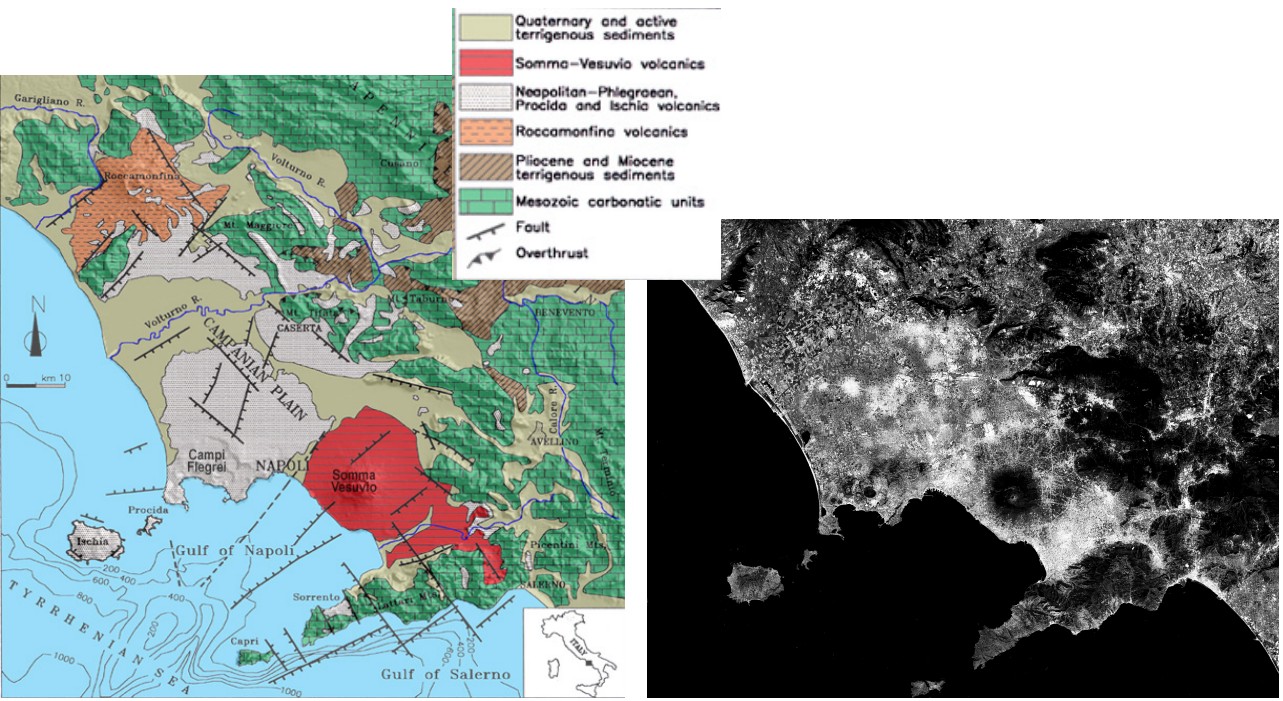

**Fig.**

**Figure 3:** Geological sketch map of the Campanian Plain, with the Neapolitan volcanic area (left, Modified from Orsi et al., 1996). Also shown (right, modified from Google Earth) is a map with the most densely urbanized areas evidenced (more intense white means denser urbanization). Note than more than 3 million people live in the shown volcanic area, than making it the most risky in the World.


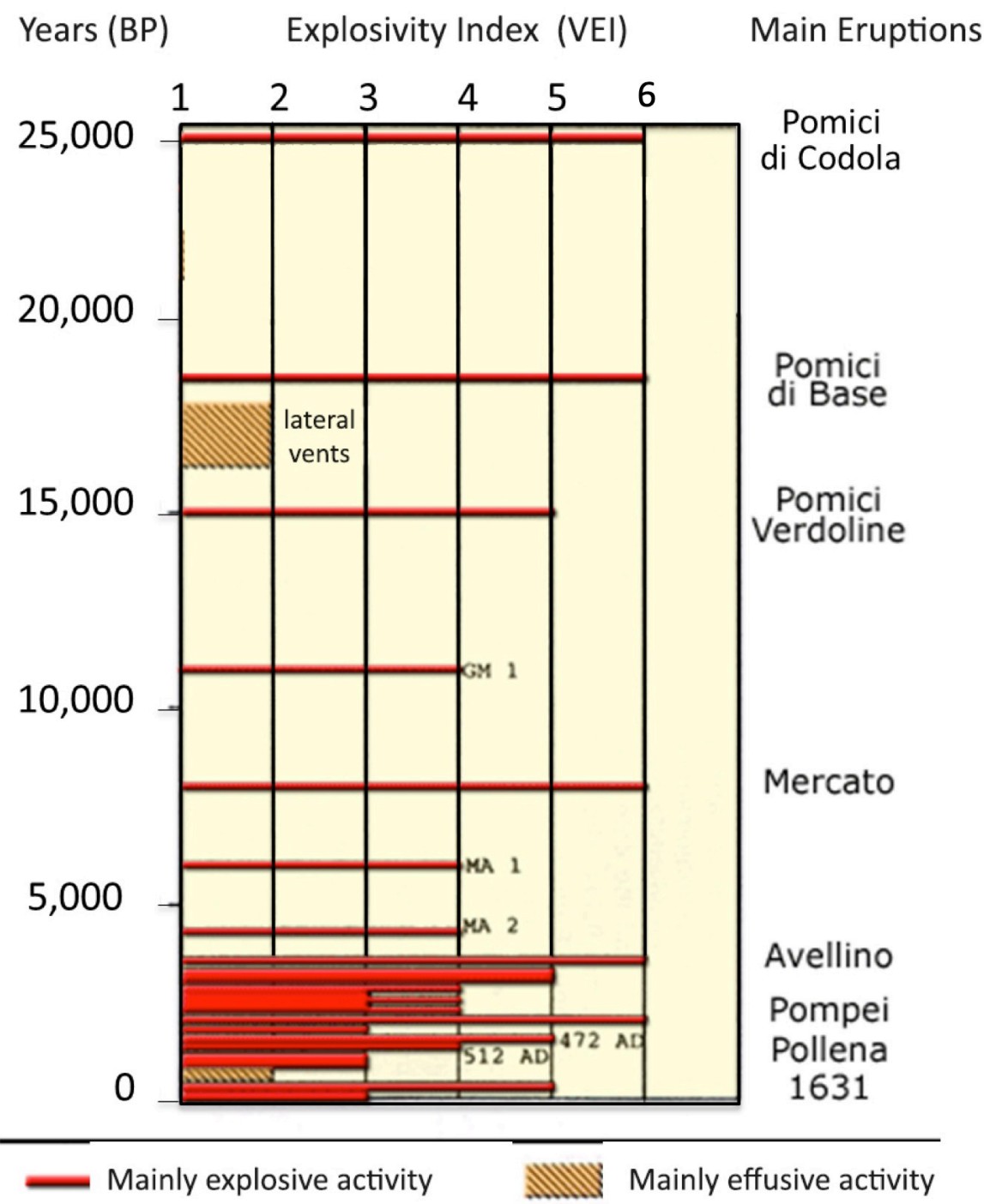

**Figure 4:** Eruptive history at Somma-Vesuvius volcano (redrawn after Osservatorio Vesuviano-INGV website www.ov.ingv.it and Rolandi, 1998).

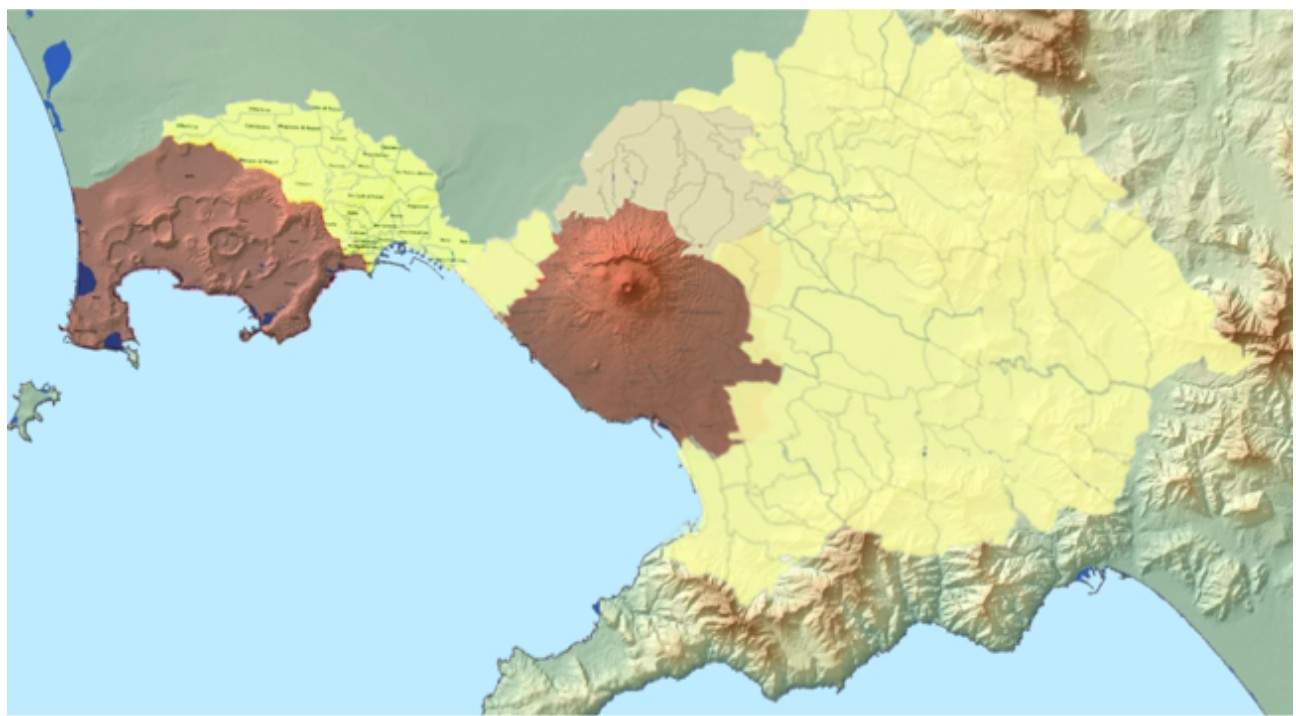

**Figure 5:** Red and Yellow zones for Campi Flegrei and Vesuvius. Red zones are the areas with maximum pyroclastic flow hazard, which have to be completely evacuated within 72 hours from the declaration of 'Red alert'. Yellow zones are the areas with maximum hazard for accumulation of pyroclastic falls on flat roofs (visit http://www.protezionecivile.gov.it/risk-activities/volcanic-risk/italian-volcanoes)


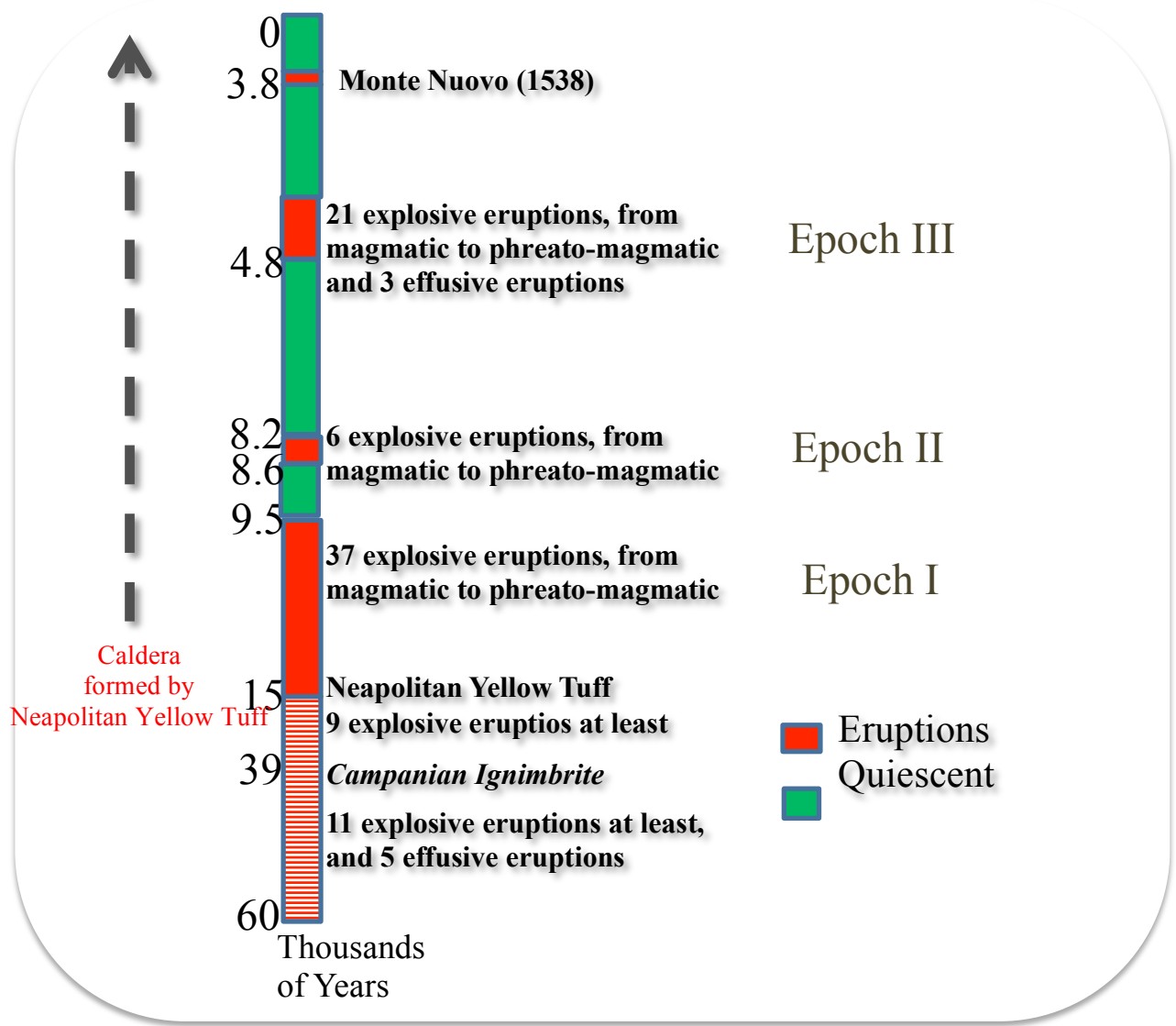

**Figure 6:** Eruptive history of Campi Flegrei caldera. Note that, according to recent results (De Natale et al., 2016; Rolandi et al., 2019b) the Campi Flegrei caldera was formed by Neapolitan Yellow Tuff (15 ky BP), and only hosted secondary events of the Campanian Ignimbrite (39 ky BP) whose main vents opened North of Campi Flegrei area (in the Campanian Plain).


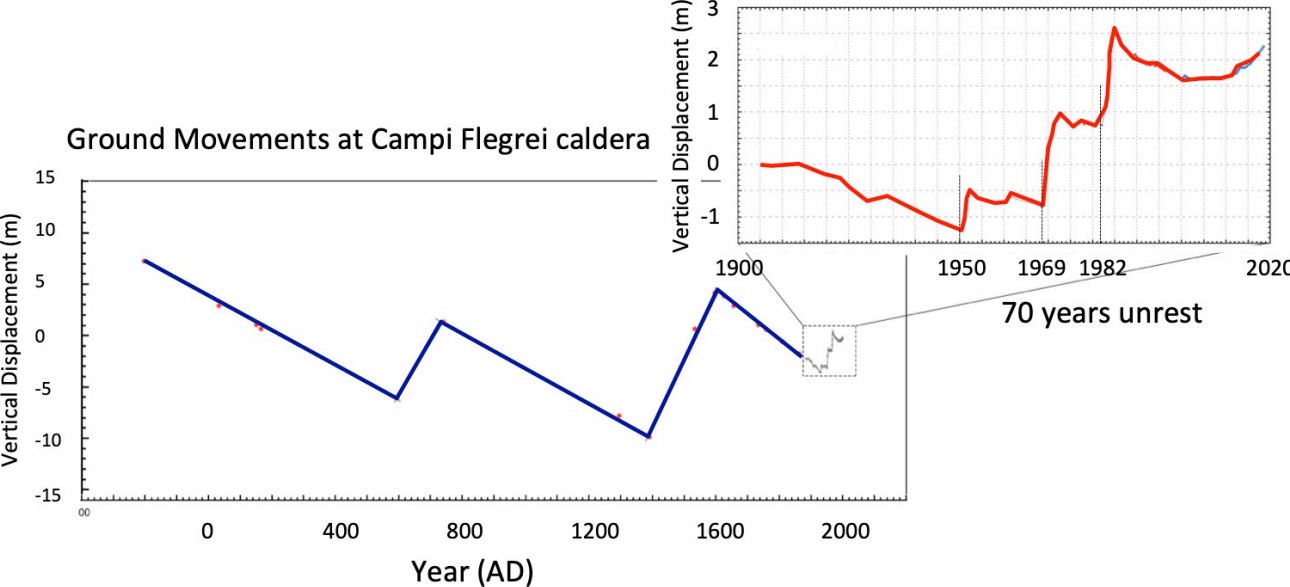

**Figure 7:** Ground movements at the Campi Flegrei caldera centre (Port of Pozzuoli town). The larger figure shows secular displacements, inferred from the traces of marine ingression on the archaeological remains of Serapeo (redrawn after Bellucci et al., 2006 and Troise et al., 2019). The exploded figure shows the vertical displacements observed since 1900 to present; data here are from precision levellings until year 2000, and then from continuous GPS measurement.

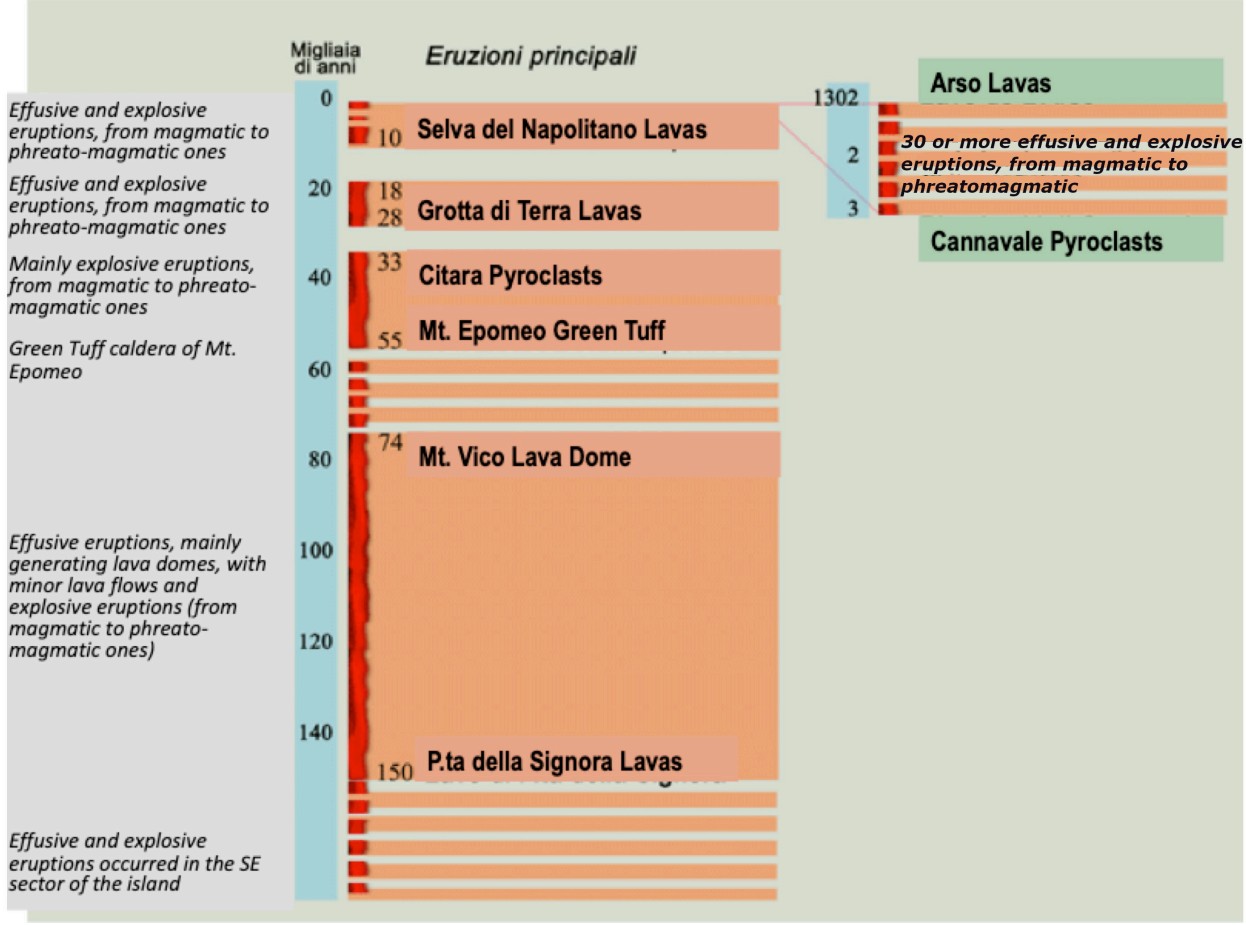

**Figure 8:** Eruptive history of Ischia island