# Peer review of "The Volcanoes of Naples: how effectively mitigating the highest volcanic risk in the World?"

_Natural Hazards and Earth System Sciences, 2020_

## Short Comment (SC1) · 14 Apr 2020

The paper is extremely interesting. It is mainly a review paper (up to line 360) that correctly points out both the enormous problems in forecasting (worldwide) and in emergency management (in the Napoli area). The last part suggests a very interesting idea, that of incentivating people to move. If i remember correctly however the italian authorities already tried to do so in the Vesuvius area in the 90s but they failed. I think the authors should address or at least mention this issue. Two more minor remarks: 1) I would like to see a figure of pozzuoli uplift, since it is very hard to follow description 2) I would like a brief discussion on Roccamonfina (the fourt volcanic apparatus in the area) as well as the fact that the ignimbrites sees not ot originate by any of the kown volcanoes in the area (if i remember corretly)

Interactive
comment

---

## Author Comment (AC1) · 15 Apr 2020

Dear Colleague,

we thank you for your constructive comments. We agree with your suggestion to discuss a little more the framework of the Campanian Plain and the role of Roccamonfina volcano. Actually, we are accurately studying the volcanism in the Campanian Plain, in a new perspective which considers such a whole area as the most volcanologically important and active, rather than the single volcanoes which erupted in recent times. Roccamonfina is in fact, in our opinion, just the northernmost boundary of such a larger volcanic area. We will also add, just as you suggest, more details (in the figure) about the uplift episodes occurring at Campi Flegrei, which is the area most concerning, at

present, giving the long lasting unrest.

Best Regards

---

## Short Comment (SC2) · 22 May 2020

Following an overview of the historic behaviour of volcanoes in the Neapolitan district, this paper highlights the importance of planning for the long-term - even permanent - displacement of several hundred thousand people following a major volcanic emergency. The review cites the authors' previous work, but could be enhanced by adding references to studies by other groups (see suggestions at end).

Previous discussions of the emergency plans, especially for Vesuvius, have focussed on preparations before an eruption (e.g., Rolandi (2010) J Volcanol Geotherm Res, 189, 347-362; Papale (2017) J App Volcanol 6, 13). This paper raises the importance of considering also how to mitigate the economic and social consequences of long-

term evacuation, which it has identified as a key goal for interdisciplinary social and volcanological studies. The results will have implications beyond the Neapolitan area. Nearly 70 cities with populations exceeding 100,000 live with the threat from volcanic eruptions (Heiken, 2013, Dangerous neighbors: volcanoes and cities. Cambridge) and so effective measures for permanently reducing the numbers at risk will have value worldwide.

The paper also notes the advantages of evacuation by sea (Lines 335-340). Have the authors assessed the numbers that could realistically be transported, especially during the 72 hours of the emergency evacuation? Local ferries have a notional capacity of 1,000 passengers and so c. 100 journeys would be required to transport numbers on the order of 100,000. For ferries to make a significant contribution, might it be necessary to increase the official evacuation time to more than 72 hours?

Fig. 4. shows an age of about 18 ka BP for the Pomici di Base at Somma-Vesuvius, whereas the text (Line 131) quotes 22 ka BP. Could the preferred age be confirmed? Rolandi (1998) has also recognised an earlier plinian eruption - the Codola eruption - at 25,000 BP (Rolandi, G., 1998. The eruptive history of Somma–Vesuvius. In: Cortini, M., De Vivo, B. (Eds.), Volcanism and archaeology in Mediterranean area. Research Signpost, 3, 77–88. He may additionally have described this eruption in a more widely distributed journal).

Fig. 7. The graph (and associated analysis) of ground movements at Pozzuoli, in Campi Flegrei, since Roman times is due to Bellucci et al. (2006, Geol Soc London, Spec Pub, 269, 141–158) and not to Troise et al. (2018) nor to Morhange et al. (2006 - Line 186).

Examples of additional studies that could be cited to broaden the review include:

(1) At Vesuvius, probabilistic analysis of hazard and risk have also been presented by Scandone et al. (1993, J Volcanol Geotherm Res 58, 263-271), Neri et al. (2008, J Volcanol Geotherm Res 178, 397-415), and Lirer et al. (2010, Bull Volcanol (2010) 72,

411-429).

(2) At Campi Flegrei, alternative reconstructions of ground movements since Roman times have been proposed by Parascandola (1947, I Fenomeni Bradisismici del Serapeo di Pozzuoli. Genovese, Naples); Dvorak. & Mastrolorenzo (1991, USGS Spec Paper, 263, 1-47); Morhange et al. (2006, Geology, 34, 93-96); and Di Vito et al. (2016, Sci. Rep. 6, 32245).
* * *

---

## Referee Comment (RC1) · Roberto Moretti (Referee) · 23 May 2020

Review of the manuscript " The Volcanoes of Naples: how effectively mitigating the highest volcanic risk in the World? " by Giuseppe De Natale, Claudia Troise, Renato Somma.

This is a very nice article that makes the point about a hot topic such as volcanic risk in the Neapolitan area. I have a few comments to improve the paper and make it a real reference for similar summaries about volcanic risk:

1) I would spend words to distinguish between long-term and short-terms assessment; This is particularly about hazard, and it is about forecasting during unrest. At Campi Flegrei caldera this is an even hotter topic. More in general, I think the paper would

benefit of this: the nice introduction seems to prelude to some discussion of the short-term forecasting, especially when false and missing alarms are described or where it is said that successful decision where taken "in progress" (e.g., Hekla or Montserrat ). So a brief description of the state-of-the-art, including contradictions, can be given, very likely when describing each of the three Neapolitan volcanoes. Of course, most of the interest turns around Campi Flegrei and its ongoing unrest. This has a peculiar relevance, given the high-level of OV monitoring and the impressive amount of publications that have appeared based on monitoring data.

2) The concept of progressive evacuation is important. Phasing is already invoked for other emergency plans (e.g. the La Soufrière the Guadeloupe one approved by Préfecture de Guadeloupe in France). It would be highly interesting if more insights and/or point of views could be given for CF caldera, where the main vent of next eruption is known probabilistically and where a robust local phreatic phase could start anticipating the magmatic one, which in turn can evolve following different scenarios. I think that the concept of phasing/progressive evacuation might already be introduced around line 330, where the logistical non-sense of an immense red area is discussed.

Some specific comments: -Line 60 "as it can be"...

-Personally I think that entire cycles of public conferences should be given around the concepts of "false" and "missing" alarms. This is likely one of the best way to make people understanding and appreciating the uncertainty of (short-term) hazard assessment, eruption forecasting and decision-making. It implies, of course, that whatever one may do in hazard assessment and forecasting is likely to be wrong for someone else. I think that in the case of many codes of law those two concepts may lead to opposite juridical implications and force a priori the decision (think about the "procurato allarme" and "mancato allarme"). I wonder if the Authors wants to spend few words on this.

-Line 140, about the De Natale et al; (2000) interpretation: please say few words on

the explanation offered in that paper.

-Line 190: I think that the reference here should be also given to Moretti, R., Troise, C., Sarno, F., & De Natale, G. (2018). Caldera unrest driven by CO2-induced drying of the deep hydrothermal system. Scientific reports, 8(1), 1-11. In this paper the symmetry between post-1984 subsidence and the on-going unrest is described for the first time.

-Line 206: there is place to cite papers that promote this hypothesis: Moretti et al 2017 G3, Moretti et al., 2018 SciRep, Troise et al. 2018, but also Moretti et al. 2013 (on EPSL) where it was first formulated. The paper that better summarizes intepretations and querelles is certainly "Moretti, R., De Natale, G., & Troise, C. (2020). Hydrothermal versus magmatic: geochemical views and clues into the unrest dilemma at Campi Flegrei. In Vesuvius, Campi Flegrei, and Campanian Volcanism (pp. 371-406). Elsevier. Âż Again, a brief description of the short-term hazard assessement (i.e. outcomes of monitoring quantities) could help, especially for CFc and its unrest.

-Line 345 on: people reallocation and 2nd life is a really good point of discussion Is any previous experience about this ? perhaps from different experiences such as cyclones. If yes, please cite.

-Line 355 on. Please take it as a very minor point that you can obviously disregard: could you do some parallel with the economic impact of the covid19 pandemics ? I say that because it would help a lot in terms of perception.

-Line 445 on. Still about 2nd life. Given the "size" of the problem you outline, I wonder if this could be part of a general socio-economic development plan of Keynesian nature at a national scale. Again, do you know if similar experiences have been done in this sense, even at a smaller scale and for different risks ?

---

## Short Comment (SC3) · 6 Jun 2020

Scientific Significance: 1 (excellent) Does the manuscript represent a substantial contribution to the understanding of natural hazards and their consequences (new concepts, ideas, methods, or data)? YES

Scientific Quality: 1 (excellent) Are the scientific and/or technical approaches and the applied methods valid? Are the results discussed in an appropriate and balanced way (clarity of concepts and discussion, consideration of related work, including appropriate references)? YES

Presentation Quality: 1 (excellent) Are the scientific data, results and conclusions presented in a clear, concise, and well-structured way (number and quality of fig-

ures/tables, appropriate use of technical and English language, simplicity of the language? YES

Suggestion: publish with very minor revision

Review of the manuscript " The Volcanoes of Naples: how effectively mitigating the highest volcanic risk in the World? " by Giuseppe De Natale, Claudia Troise, Renato Somma.

The paper presents a clear review of the heavy problem of volcanic risk mitigation in the Neapolitan area. The paper starts recalling the main problems actually involved in the eruption forecast, which, according to the most recent literature, has a very low percentage of successes. Starting from such consideration, the authors analyse the main features, eruptive history and hazard of each one of the three Neapolitan volcanoes, and then proceeds to analyse the main problems to design an effective Emergency Plan, which is really feasible from a logistic, economic and social point of view. While describing the optimal features of a realistic Emergency Plan, the authors clearly put in evidence the limits and problems of the present Emergency Plans existing for Vesuvius and Campi Flegrei. The resulting framework is an innovative one, very useful not only for this extremely populated area, but also for any other populated one, prone to volcanic risk in the World.

The paper is surely of high interest for the journal, and for the volcanological research applied to risk mitigation. It is generally well written (I don't make any language correction, because not of English mother tongue), with all the main concepts well explained. The conclusions are well supported by data, literature and volcanological considerations. Figures are all necessary and well understandable.

I surely support publication, almost in the present form, and give some suggestions for minor revisions, the authors should consider to include: 1) the authors should include, in the references about the discussion on the Vesuvius emergency plans, the paper by Rolandi (2010); 2) Regarding fig.7, the authors should mention the paper by Bellucci

et al. (2006) which, at my knowledge, has been the first one to propose the depicted behaviour for the secular ground movements; 3) You could perhaps spend some more lines explaining the benefits of a 'progressive evacuation' approach, which I find absolutely correct as opposite to a 'giant' red zone to suddenly evacuate in few days. 4) You could explain a little more the model for background seismicity at Vesuvius, where you quote De Natale et al., 2000.

Best Regards

Giuseppe Rolandi

---

## Referee Comment (RC2) · Marco Sacchi (Referee) · 10 Jun 2020

I have found this manuscript a well-constructed piece of work in which the authors address a review of the structure and recent evolution of the active volcanoes in the Neapolitan district, integrated with an interesting in-depth discussion on the several issues associated with the long-debated planning for the mitigation of volcanic risk, eruption forecast and alerts in the area (and in active volcanic areas in general).

The review of the volcanological framework of the Neapolitan district is concise but informative and it may eventually benefit from (even very short) additional mention/discussion on Roccamonfina, as the NW boundary of the Campania volcanic area, as well as on the volcanic edifices and subvolcanic structures offshore the Napoli Bay,

as the S border of the volcanic district, (see detailed reference list below). The mentioning of the (significantly vast) submerged volcanic features of the Napoli Bay may also serve to highlight the importance the volcanic hazard associated with the (relatively less known) submarine volcanic features for a more comprehensive and reliable hazard estimate and mapping.

The analysis on emergency plans and evacuation plans, along with strength and weakness of the two components in the play of the volcanic risk mitigation is well introduced and discussed in sections 4 and 5. The most relevant and innovative outcome of the paper is probably the recognized importance of setting up a fully integrated, flexible risk mitigation strategy including three major components: 1) a sustainable long term plan to decrease population living in the area; 2) a long-term policy to enhance and diversify the regional transportation network and escape routes (i.e. terrestrial, marine, aerial?) in case eruption alert; 3) a sustainable long-term integration of the population progressively leaving from the volcanic area into the social and economic network of the hosting region(s).

All in all, I suggest the manuscript be accepted for publication with very minor revision. Here's a list of further detailed comments/suggestions:

line 144-145: Consider the possibility of adding (a selection of) the following references: "The main volcanic hazards are pyroclastic flows and ash/pumice fallout (e.g. Sacchi et al., 2005; 2019; 2020), but also associated hazards like earthquakes, lahars, lava flows and floods (Sacchi et al., 2009), need to be considered".

line 177: Cassano and La Torre, 1987 instead of "Cassano et al., 1987"

line 177: Capuano and Achauer, 2003 instead of "Capuano et al., 2003"

line 177-178: Consider the possibility of adding (a selection of) the following references: "has a radius of about 3 km, with center approximately located at the Pozzuoli town harbour" (Sacchi et al., 2014; Somma et al., 2015; Steinmann et al, 2016; 2018 Sacchi

et al, 2019; 2020).

Line 192: Consider the possibility of adding the following reference: "(the so-called 'mini-uplift' episodes: see Gaeta et al., 2003; Troise et al., 2007;" Iuliano et al., 2015).

Line 228-229: Consider the possibility of adding the following reference: Ischia island, located South-West of Campi Flegrei, is another volcanic field, characterized by both effusive and explosive eruptions (Passaro et al., 2015).

List of suggested References:

Sacchi M., Insinga D., Milia A., Molisso F., Raspini A., Torrente M.M., Conforti A., 2005. Stratigraphic signature of the Vesuvius 79 AD event off the Sarno prodelta system, Naples Bay. Marine Geology 222–223 (2005) 443– 469.

Sacchi M., Molisso F., Esposito E., Insinga D., Lubritto C., Porfido S., Tóth T., Violante C., 2009. Insights into flood-dominated fan deltas: very high-resolution seismic examples off the Amalfi cliffed coasts, Eastern Tyrrhenian Sea. In: Violante, C. (ed.) Geohazard in Rocky Coastal Areas. The Geological Society, London, Special Publications, 322, 33–71.

Sacchi M., Pepe F., Corradino M., Insinga D.D., Molisso F., Lubritto C., 2014. The Neapolitan Yellow Tuff caldera offshore the Campi Flegrei: Stratal architecture and kinematic reconstruction during the last 15 ky. Marine Geology, 354, 15-33.

Iuliano S., Matano F., Caccavale M., Sacchi M., 2015. Annual rates of ground deformation (1993–2010) at Campi Flegrei, Italy, revealed by Persistent Scatterer Pair (PSP) – SAR interferometry. International Journal of Remote Sensing, vol. 36, No. 24, 6160–6191.

Passaro S., de Alteriis G., Sacchi M., 2016. Bathymetry of Ischia Island and its offshore (Italy), scale 1:50.000. Journal of Maps, Vol. 12, No. 1, 152–159, doi: 10.1080/17445647.2014.998302.

Passaro S., Tamburrino S., Vallefuoco M., Tassi F., Vaselli O., Giannini L., Chiodini G., Caliro S., Sacchi M., Rizzo A.L., Ventura G., 2016. Seafloor doming driven by degassing processes unveils sprouting volcanism in coastal areas. Scientific Reports, 6, 22448; doi: 10.1038/srep22448.

Somma R., Iuliano S., Matano F., Molisso F., Passaro S., Sacchi M., Troise C., De Natale G., 2016. High-resolution morpho-bathymetry of Pozzuoli Bay, southern Italy. Journal of Maps, Vol. 12, No. 2, 222–230, doi:10.1080/17445647.2014.1001800.

Steinmann L., Spiess V., Sacchi M., 2016. The Campi Flegrei caldera (Italy): Formation and evolution in interplay with sea-level variations since the Campanian Ignimbrite eruption at 39 ka. Journal of Volcanology and Geothermal Research, 327, 361-374.

Steinmann L., Spiess V., Sacchi M., 2018. Post-collapse evolution of a coastal caldera system: Insights from a 3D multichannel seismic survey from the Campi Flegrei caldera (Italy). Journal of Volcanology and Geothermal Research, 349, 83-98.

Sacchi M., De Natale G., Spiess V., Steinmann L., Acocella V., Corradino M., de Silva S., Fedele A., Fedele L., Geshi N., Kilburn C., Insinga D., Jurado M-J., Molisso F., Petrosino P., Passaro S., Pepe F., Porfido S., Scarpati C., Schmincke H-U., Somma R., Sumita M., Tamburrino S., Troise C., Vallefuoco M, Ventura G., 2019. A roadmap for amphibious drilling at the Campi Flegrei caldera: insights from a MagellanPlus workshop. Scientific Driling, 7, 1–18; doi: 10.5194/sd-7-1-2019.

Sacchi M., Passaro S., Molisso F., Matano F., Steinmann L., Spiess V., Pepe F., Corradino M., Caccavale M., Tamburrino S., Esposito G., Vallefuoco M., Ventura G., 2020. The Holocene marine record of unrest, volcanism, and hydrothermal activity of Campi Flegrei and Somma Vesuvius. In: B. De Vivo, H.E. Belkin and G. Rolandi (Eds.) Vesuvius, Campi Flegrei, and Campanian Volcanism, Elsevier Inc., Amsterdam, l435-469.

---

## Author Comment (AC2) · 10 Jun 2020

Answers to Short Comment 2 (Christopher R.J. Kilburn)

We thank the colleague for the very helpful comments and suggestions. We try to synthesize our answers to the comments/suggestions received, since it is not a formal review.

'Previous discussions of the emergency plans, especially for Vesuvius, have focussed on preparations before an eruption (e.g., Rolandi (2010) J Volcanol Geotherm Res, 189, 347-362; Papale (2017) J App Volcanol 6, 13). This paper raises the importance of considering also how to mitigate the economic and social consequences of long-erm evacuation, which it has identified as a key goal for interdisciplinary social and

volcanological studies. The results will have implications beyond the Neapolitan area. Nearly 70 cities with populations exceeding 100,000 live with the threat from volcanic eruptions (Heiken, 2013, Dangerous neighbors: volcanoes and cities. Cambridge) and so effective measures for permanently reducing the numbers at risk will have value worldwide.'

Answer: Thank you, we added all he suggested references in the revised version.

The paper also notes the advantages of evacuation by sea (Lines 335-340). Have the authors assessed the numbers that could realistically be transported, especially during the 72 hours of the emergency evacuation? Local ferries have a notional capacity of 1,000 passengers and so c. 100 journeys would be required to transport numbers on the order of 100,000. For ferries to make a significant contribution, might it be necessary to increase the official evacuation time to more than 72 hours?

Answer: As we specified, in one case at least, namely Ischia island, the evacuation by sea is the only possibility; in this case, the people to evacuate would be reasonably less than the whole island population (there is not an emergency plan or red zone yet), i.e. about 60,000 people. However, although local ferries capacities are more limited, large cruise ships can host more than 3000, 4000 or else 5000 people. So, little more than 100 trips to the closest safe places could be enough to evacuate even the largest red zones. Obviously, the departure ports should be able to host the large cruise ships. This also calls for a serious improvement of the port basins and of their facilities, which fall into the more general infrastructural improvement needed in the areas of maximum volcanic hazard.

Fig. 4. shows an age of about 18 ka BP for the Pomici di Base at Somma-Vesuvius, whereas the text (Line 131) quotes 22 ka BP. Could the preferred age be confirmed? Rolandi (1998) has also recognised an earlier plinian eruption - the Codola eruption - at 25,000 BP (Rolandi, G., 1998. The eruptive history of Somma–Vesuvius. In: Cortini, M., De Vivo, B. (Eds.), Volcanism and archaeology in Mediterranean area. Research

[Figure]

Signpost, 3, 77–88. He may additionally have described this eruption in a more widely distributed journal).

Answer: Thank you for having pointed out the mistake, and for suggesting to include Codola eruption. The most accredited age in literature, for the Pomici di Base eruption, is 18 ky BP, just as shown in the figure. So, we corrected the age at lin 131. Furthermore, we did not mention the Codola 25 ky BP eruption, which is however now recognized by most of the recent literature. We then also added it in the figure, which was redrawn from the website of INGV-Osservatorio Vesuviano, and did not report the Codola eruption.

Fig. 7. The graph (and associated analysis) of ground movements at Pozzuoli, in Campi Flegrei, since Roman times is due to Bellucci et al. (2006, Geol Soc London, Spec Pub, 269, 141–158) and not to Troise et al. (2018) nor to Morhange et al. (2006 - Line 186).

Answer: Thank you, we sincerely apologise to have missed this reference, which is undoubtedly the first one in which such a secular deformation behaviour was proposed (it was noted also by the reviewer 2). We added the original reference to the revised manuscript.

Examples of additional studies that could be cited to broaden the review include: (1) At Vesuvius, probabilistic analysis of hazard and risk have also been presented by Scandone et al. (1993, J Volcanol Geotherm Res 58, 263-271), Neri et al. (2008, J Volcanol Geotherm Res 178, 397-415), and Lirer et al. (2010, Bull Volcanol (2010) 72, 411-429). (2) At Campi Flegrei, alternative reconstructions of ground movements since Roman times have been proposed by Parascandola (1947, I Fenomeni Bradisismici del Serapeo di Pozzuoli. Genovese, Naples); Dvorak. & Mastrolorenzo (1991, USGS Spec Paper, 263, 1-47); Morhange et al. (2006, Geology, 34, 93-96); and Di Vito et al. (2016, Sci. Rep. 6, 32245).

Answer: Thank you. We knew such papers, and not quoted all of them just to avoid to

make references too much cumbersome. However, we actually included some of them.

Please also note the supplement to this comment:
https://www.nat-hazards-earth-syst-sci-discuss.net/nhess-2020-51/nhess-2020-51-AC2-supplement.pdf

---

## Author Comment (AC3) · 10 Jun 2020

Answers to the Reviewer 1 (Roberto Moretti)

We thank the reviewer for his helpful suggestions; we are going here to answer point by point:

1) I would spend words to distinguish between long-term and short-terms assessment; This is particularly about hazard, and it is about forecasting during unrest. At Campi Flegrei caldera this is an even hotter topic. More in general, I think the paper would benefit of this: the nice introduction seems to prelude to some discussion of the short-term forecasting, especially when false and missing alarms are described or where it is said that successful decision where taken "in progress" (e.g., Hekla or Montserrat ). So

a brief description of the state-of-the-art, including contradictions, can be given, very likely when describing each of the three Neapolitan volcanoes. Of course, most of the interest turns around Campi Flegrei and its ongoing unrest. This has a peculiar relevance, given the high-level of OV monitoring and the impressive amount of publications that have appeared based on monitoring data.

Answer: we thank the reviewer for such an interesting suggestion. We therefore added a little more discussion about the actual difficulty to distinguish real precursors, mainly at Campi Flegrei caldera where somewhat anomalous activity associated to the unrest lasts since many decades.

2) The concept of progressive evacuation is important. Phasing is already invoked for other emergency plans (e.g. the La Soufrière the Guadeloupe one approved by Préfecture de Guadeloupe in France). It would be highly interesting if more insights and/or point of views could be given for CF caldera, where the main vent of next eruption is known probabilistically and where a robust local phreatic phase could start anticipating the magmatic one, which in turn can evolve following different scenarios. I think that the concept of phasing/progressive evacuation might already be introduced around line 330, where the logistical non-sense of an immense red area is discussed.

Answer: thank you, we agree the concept of progressive evacuation is absolutely important, and should be necessarily included in any emergency plan for Neapolitan vlcanoes. In fact, it is non realistic to think to evacuate in few days 600,000-700,000 people, given the very high probability of false alarm even in presence of macroscopic 'anomalies' of seismicity, ground deformation, geochemistry of gases and waters, and any other signal. Actually, the two evacuations of 1970 and 1984 implicitly followed the concept of 'progressive evacuation', because only the areas considered at highest immediate risk in case of eruption were involved, not excluding further evacuation of larger areas. We have followed the reviewer's suggestion, and introduced the concept of progressive evacuation in the part where we discuss the problems of defining a very large red zone.

Answers to specific comments follow:

-Line 60 "as it can be". . . -Personally I think that entire cycles of public conferences should be given around the concepts of "false" and "missing" alarms. This is likely one of the best way to make people understanding and appreciating the uncertainty of (short-term) hazard assessment, eruption forecasting and decision-making. It implies, of course, that whatever one may do in hazard assessment and forecasting is likely to be wrong for someone else. I think that in the case of many codes of law those two concepts may lead to opposite juridical implications and force a priori the decision (think about the "procurato allarme" and "mancato allarme"). I wonder if the Authors wants to spend few words on this.

Answer: thank you, this is a very intriguing question, which rightly involves also juridical aspects, which can become more and more important in natural disaster management. In Italy, we have a very good example of that in the case of 2009 L'Aquila earthquake. This is a further, very good reason why the very hard problem of volcanic risk management in the Neapolitan area has to be afforded with maximum transparency and rationality, without understating any of the numerous problems involved. This very important question is the background to the paper content. However, we agreed to discuss even more this point, in the revised version.

-Line 140, about the De Natale et al; (2000) interpretation: please say few words on the explanation offered in that paper.

Answer: Ok, we added some more explanation of that. This request is common to the one from reviewer 1. Thank you.

-Line 190: I think that the reference here should be also given to Moretti, R., Troise, C., Sarno, F., & De Natale, G. (2018). Caldera unrest driven by $CO_2$-induced drying of the deep hydrothermal system. Scientific reports, 8(1), 1-11. In this paper the symmetry between post-1984 subsidence and the on-going unrest is described for the first time.

Answer: Yes, of course. We completely agree and included that reference.

-Line 206: there is place to cite papers that promote this hypothesis: Moretti et al 2017 G3, Moretti et al., 2018 SciRep, Troise et al. 2018, but also Moretti et al. 2013 (on EPSL) where it was first formulated. The paper that better summarizes intepretations and querelles is certainly "Moretti, R., De Natale, G., & Troise, C. (2020). Hydrothermal versus magmatic: geochemical views and clues into the unrest dilemma at Campi Flegrei. In Vesuvius, Campi Flegrei, and Campanian Volcanism (pp. 371-406). Elsevier. Âz ÌĞ Again, a brief description of the short-term hazard assessement (i.e. outcomes of monitoring quantities) could help, especially for CFc and its unrest.

Answer: Thank you for the suggestions. We included all of them in the revision.

-Line 345 on: people reallocation and 2nd life is a really good point of discussion Is any previous experience about this ? perhaps from different experiences such as cyclones. If yes, please cite.

Answer: It is a very interesting question. At our knowledge, there has not been a similar experience till now, regarding programmed evacuation to avid a disaster in a very hazardus area. There has been experience of relocation after a disaster (there are some experience of partial relocation of population also in Italy, after large tectonic earthquakes). We found a good example of complete relocation of a small town (900 people): Valmeyer, Illinois, where population, after a catastrophic flood of the Mississipi in 1993, moved the whole own at a new site in 1995 (Rozdilsky J. Environment and Planning Newsletter. Indianapolis, IN: Environmental, Natural Resources and Energy Division, American Planning Association, Center for Urban Policy and the Environment, School of Public and Environmental Affairs, Indiana University – Purdue University; 1996. Flood-related relocation of Valmeyer: Implications for the development of sustainable cities.). Kiruna, Sweden, 23,000 inhabitants, is another town which is going to be moved 3 km apart, because of hazard posed by ore activities which are causing continuing ground sinking and felt seismicity (Dineva, S & Boskovic, M 2017,

'Evolution of seismicity at Kiruna Mine', in J Wesseloo (eds.), Proceedings of the Eighth International Conference on Deep and High Stress Mining, Australian Centre for Geomechanics, Perth, pp. 125-139, https://doi.org/10.36487/ACG_rep/1704_07_Dineva).

-Line 355 on. Please take it as a very minor point that you can obviously disregard: could you do some parallel with the economic impact of the covid19 pandemics ? I say that because it would help a lot in terms of perception.

Answer: Ok, it is an intriguing question. Obviously, when we submitted the paper there was not such experience. However, we have now included a small discussion on what we could learn from the pandemic, useful for our problem.

-Line 445 on. Still about 2nd life. Given the "size" of the problem you outline, I wonder if this could be part of a general socio-economic development plan of Keynesian nature at a national scale. Again, do you know if similar experiences have been done in this sense, even at a smaller scale and for different risks ?

Answer: At our knowledge, there has not been a similar experience till now, at las on large economic scale

We hope to have now satisfied all the reviewer concerns.

Best Regards

Giuseppe De Natale, Claudia Troise, Renato Somma

Please also note the supplement to this comment:
https://www.nat-hazards-earth-syst-sci-discuss.net/nhess-2020-51/nhess-2020-51-AC3-supplement.pdf

---

## Author Comment (AC4) · 10 Jun 2020

Answers to the Reviewer 2 (Listed as SC3: Giuseppe Rolandi)

We thank the reviewer for his helpful suggestions; we are going here to answer point by point:

1) The authors should include, in the references about the discussion on the Vesuvius emergency plans, the paper by Rolandi (2010);

Answer: we agree, and apologise we did not before. However, the reference has been added in the revised version of the manuscript.

2) Regarding fig.7, the authors should mention the paper by Bellucci et al. (2006)

[Figure]

which, at my knowledge, has been the first one to propose the depicted behaviour for the secular ground movements;

Answer: we agree, we missed to mention it in the first version, but we added in the revised one.

3) You could perhaps spend some more lines explaining the benefits of a 'progressive evacuation' approach, which I find absolutely correct as opposite to a 'giant' red zone to suddenly evacuate in few days;

Answer: yes, we agree this is a very crucial point, because an extreme enlargment of the red zone, while giving a false feeling of safety, makes actually unlikely, or else impossible, to decide the evacuation. This is because more severe are the consequences of a false alarm (and unnecessarily evacuating 600,000-700,000 people is a really catastrophic outcome), more evidence for a really impending eruption the decision-makers (politicians) will need to decide an evacuation. But waiting more and more evidence from precursors, given the implicit, large uncertainty of their significance, can easily prove fatal. We have stressed more, in the revised version, the importance of a progressive evacuation, starting from smaller, more critical areas (just as it was made in the Pozzuoli town evacuation in 1984). Thank you for this important suggestion.

4) You could explain a little more the model for background seismicity at Vesuvius, where you quote De Natale et al., 2000;

Answer: we agree, it s an important point, and we discussed it in more detail in the revised version of the manuscript.

We hope to have now satisfied all the reviewer concerns.

Best Regards

Giuseppe De Natale, Claudia Troise, Renato Somma

Please also note the supplement to this comment:

https://www.nat-hazards-earth-syst-sci-discuss.net/nhess-2020-51/nhess-2020-51-AC4-supplement.pdf

---

## Author Comment (AC5) · 10 Jun 2020

Answers to the Reviewer 3 (Listed as RC2: Marco Sacchi)

We thank the reviewer for his helpful suggestions; we are going here to answer point by point:

The review of the volcanological framework of the Neapolitan district is concise but informative and it may eventually benefit from (even very short) additional mention/discussion on Roccamonfina, as the NW boundary of the Campania volcanic area, as well as on the volcanic edifices and subvolcanic structures offshore the Napoli Bay, as the S border of the volcanic district, (see detailed reference list below). The mentioning of the (significantly vast) submerged volcanic features of the Napoli Bay may

also serve to highlight the importance the volcanic hazard associated with the (relatively less known) submarine volcanic features for a more comprehensive and reliable hazard estimate and mapping.

Answer: We agree with this suggestion, so we included, in the overview of the Campanian Volcanic zone, more details about Roccamonfina and the submerged volcanic features of the Naples Bay. Thank you, it was a very good point.

line 144-145: Consider the possibility of adding (a selection of) the following references: "The main volcanic hazards are pyroclastic flows and ash/pumice fallout (e.g. Sacchi et al., 2005; 2019; 2020), but also associated hazards like earthquakes, lahars, lava flows and floods (Sacchi et al., 2009), need to be considered";

Answer: we agree, and have included the related references. Thank you.

line 177: Cassano and La Torre, 1987 instead of "Cassano et al., 1987" line 177: Capuano and Achauer, 2003 instead of "Capuano et al., 2003";

Answer: thank you for having pointed out this small mistakes with indicated references.

line 177-178: Consider the possibility of adding (a selection of) the following references: "has a radius of about 3 km, with center approximately located at the Pozzuoli town harbour" (Sacchi et al., 2014; Somma et al., 2015; Steinmann et al, 2016; 2018 Sacchi et al, 2019; 2020);

Answer: ok, thank you. We added some of the quoted references in the revised version.

Line 192: Consider the possibility of adding the following reference: "(the so-called 'mini-uplift' episodes: see Gaeta et al., 2003; Troise et al., 2007;" Iuliano et al., 2015);

Answer: ok, thank you. We added the suggested reference.

Line 228-229: Consider the possibility of adding the following reference: Ischia island, located South-West of Campi Flegrei, is another volcanic field, characterized by both

effusive and explosive eruptions (Passaro et al., 2015);

Answer: Ok, we added the suggested reference.

We hope to have now satisfied all the reviewer concerns.

Best Regards

Giuseppe De Natale, Claudia Troise, Renato Somma

Please also note the supplement to this comment:
https://www.nat-hazards-earth-syst-sci-discuss.net/nhess-2020-51/nhess-2020-51-AC5-supplement.pdf

―――――――――――――――――